# SpatioLM: Towards General Physical Spatial Intelligence in Vision-Language Models

**Jing Wu** [* 1]  **Jianhua Wu** [* 1 2]  **Jiayi Guan** [1]  **Jiahong Chen** [3]  **Jinghui Lu** [1]  **Hangjun Ye** [1]  **Bingzhao Gao** [2]  **Long Chen** [1]

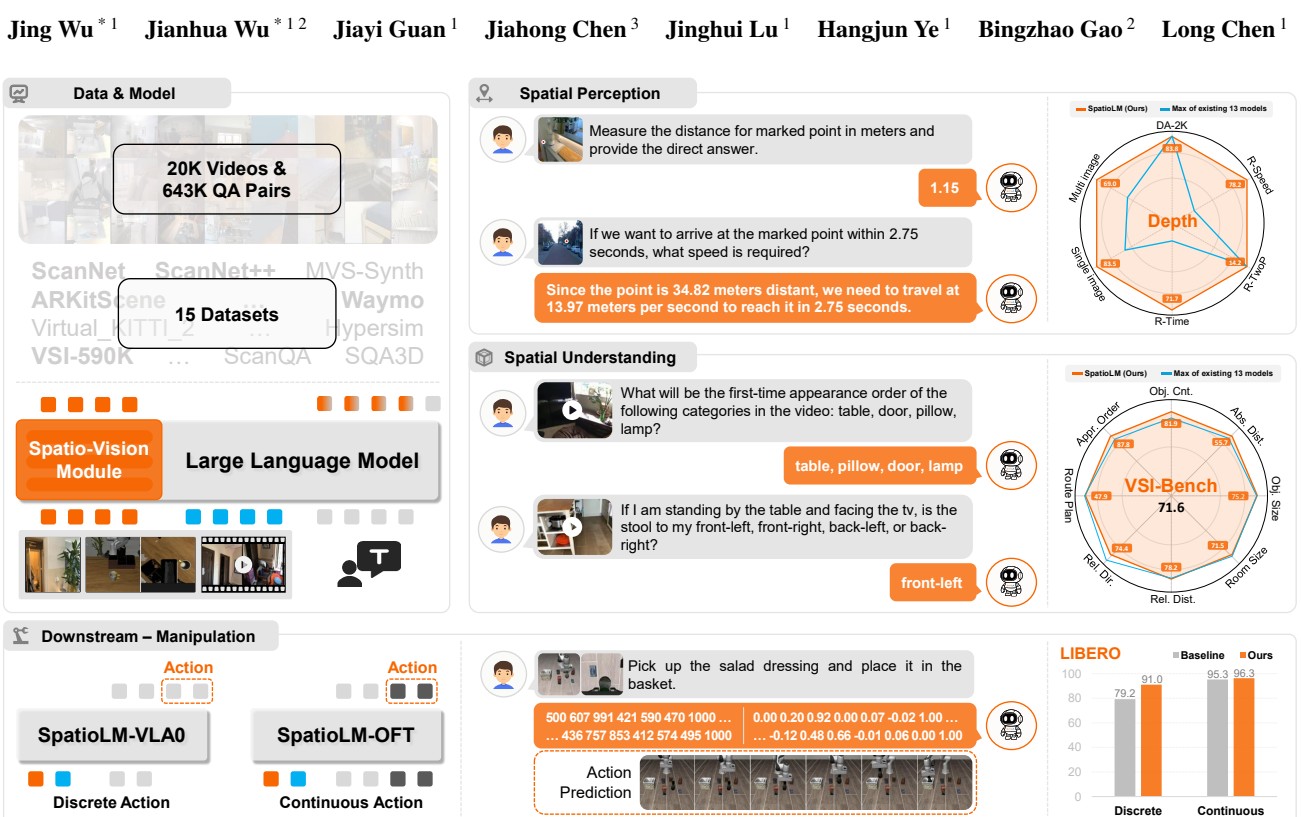

*Figure 1.* We propose *SpatioLM*, a parameter-efficient framework that improves spatial intelligence in VLMs without extra 3D prior inputs or external spatial encoders. SpatioLM achieves SOTA performance on both spatial perception and understanding tasks, and can be effectively adapted to embodied manipulation tasks under both discrete and continuous action settings.

## Abstract

Vision-Language Models (VLMs) perform well on commonsense reasoning tasks but struggle with visual spatial reasoning. Most existing solutions introduce extra 3D prior inputs or external spatial encoders, which increase complexity and degrade the underlying VLMs' general-purpose capabilities after spatial fine-tuning. To this end, we propose a parameter-efficient ***Spatio-vision Language Models (SpatioLM)***, that enhances spatial intelligence without extra 3D prior inputs or third-party spatial encoders. Concretely, we design a plug-and-play and non-invasive spatio-vision module that elicits the spatial knowledge inherent in VLMs. Furthermore, we innovatively leverage pseudo depth and camera information as supervision to guide the model in learning physically coherent representations. Extensive experiments show that SpatioLM achieves significant improvements in diverse tasks, including spatial perception and understanding while effectively limiting the degradation of general capabilities. Notably, the model achieves an impressive score of 71.6 on the VSI-Bench (the first model to surpass 70). In addition, it attains competitive performance when transferred to embodied manipulation tasks.

*Equal contribution [1]Xiaomi EV, Beijing, China [2]College of Automotive and Energy Engineering, Tongji University, Shanghai, China [3]Independent Researcher. Correspondence to: Long Chen <chenlong37@xiaomi.com>.

*Proceedings of the 43rd International Conference on Machine Learning*, Seoul, South Korea. PMLR 306, 2026. Copyright 2026 by the author(s).

# 1. Introduction

Vision-Language Models (VLMs) demonstrate exceptional semantic alignment and robust generalization (Radford et al., 2021; Liu et al., 2023b; Hurst et al., 2024; OpenAI, 2025; Comanici et al., 2025; Bai et al., 2025a) across diverse vision-language tasks, including visual question answering and multimodal reasoning. Despite significant advances in this field, constrained by the existing multimodal large-model architectures and large-scale image-text pretraining paradigm, these models generally feature rich semantic representations yet insufficient spatial reasoning. This directly results in poor performance of current VLMs on tasks requiring 3D metric reasoning and spatial understanding (Zhang et al., 2024a; Chen et al., 2025a; Qi et al., 2025; Yu et al., 2025). However, real-world scenarios, including embodied manipulation and autonomous driving, rely critically on models' capacities for spatial reasoning. Enhancing these capabilities in VLMs is therefore paramount to facilitating their real-world applications.

To enhance the intrinsic 3D metric reasoning and spatial understanding capabilities of VLMs, existing studies focusing on end-to-end model improvements have explored viable solutions from two perspectives: explicit geometric augmentation and architectural-level modification. As illustrated in Fig. 2-(a), explicit geometric augmentation methods boost the model's spatial reasoning capability by incorporating 3D prior information such as depth maps, point clouds, and camera parameters from sensors (Liu et al., 2025a; Huang et al., 2025; Zhou et al., 2025; Daxberger et al., 2025) or auxiliary estimation models (Chen et al., 2025b; Cheng et al., 2024; Liu et al., 2025b). Although effective in controlled settings, these approaches depend on specialized hardware or fragile preprocessing pipelines, limiting their scalability and robustness in open-world, RGB-only scenarios. On the other hand, Fig. 2-(b) illustrates that architectural-level modification methods externalize spatial reasoning by introducing additional spatial encoders (Ma et al., 2025; Fan et al., 2025; Zheng et al., 2025; Wu et al., 2025), rather than eliciting the intrinsic spatial knowledge of VLMs. This consequently renders such methods unable to effectively adapt to diverse spatial tasks. Furthermore, fine-tuning the VLMs on large-scale 3D data severely undermines their pretrained feature distributions, thereby inducing a significant degradation in general-purpose capabilities.

The aforementioned analysis highlights an unresolved core challenge: *how to enhance the spatial intelligence of VLMs without extra 3D prior inputs or third-party spatial encoders, while preserving their general-purpose capabilities.*

To tackle this core challenge, this work proposes a **Spatio**-vision **L**anguage **M**odel (SpatioLM), whose overall architecture is illustrated in Fig. 2-(c). Concretely, SpatioLM adopts a core paradigm of "Frozen Backbone + Plug-and-Play Spatio-Vision Module." This paradigm entails freezing all parameters of the pre-trained VLMs to preserve their general capabilities, while integrating a non-invasive Spatio-Vision Module (SV-Module) to elicit the model's inherent geometric representations. Without modifying the core structure of the base VLM, SpatioLM enables flexible activation or deactivation during the inference phase, thereby enhancing the base VLM with on-demand visual spatial reasoning capability. Furthermore, extensive experimental results demonstrate that our model achieves significant improvements in spatial perception and understanding while effectively preserving its general-purpose intelligence. Our main contributions are summarized as follows:

- We propose **SpatioLM**, a purely 2D vision-language framework that elicits implicit 3D geometric structure from pretrained VLM vision tokens, eliminating reliance on sensor-based or estimated geometric inputs.

- We innovatively design a *plug-and-play Spatio-Vision Module* that enhances visual spatial reasoning without invasive fine-tuning, effectively preventing catastrophic forgetting and maintaining VLM's generality.

- Extensive experiments demonstrate that SpatioLM outperforms existing mainstream models on tasks such as spatial perception, understanding and downstream embodied manipulation. Furthermore, the model retains favorable generalization performance via its non-invasive Spatio-Vision Module.

# 2. Related Work

We briefly overview related work here and defer our discussion of detailed related work to Appendix A.

Recent approaches to spatial perception and understanding with VLMs generally fall into three distinct paradigms. A major line of work augments VLMs with explicit geometric priors, either via camera-intrinsic normalization and depth supervision for metric perception (e.g., DepthLM (Cai et al., 2025c), SpatialBot (Cai et al., 2025a)) or through sensor-based or estimation-based 3D inputs for spatial understanding (e.g., OmniEVA (Liu et al., 2025a), 3DRS (Huang et al., 2025), RoboRefer (Zhou et al., 2025), MM-Spatial (Daxberger et al., 2025), GS-Reasoner (Chen et al., 2025b), SSR (Liu et al., 2025b)). While effective in controlled settings, these methods rely on explicit geometric inputs or multi-stage pipelines, limiting robustness and scalability in open-world RGB-only scenarios. Another line of work focuses on architectural modifications, introducing external spatial encoders (e.g., SpatialLLM (Ma et al., 2025), VLM-3R (Fan et al., 2025), VG-LLM (Zheng et al., 2025), Spatial-MLLM (Wu et al., 2025)). Although these approaches boost benchmark performance, they require invasive 3D data fine-tuning, which disrupts pretrained representations and degrades general semantic and reason-

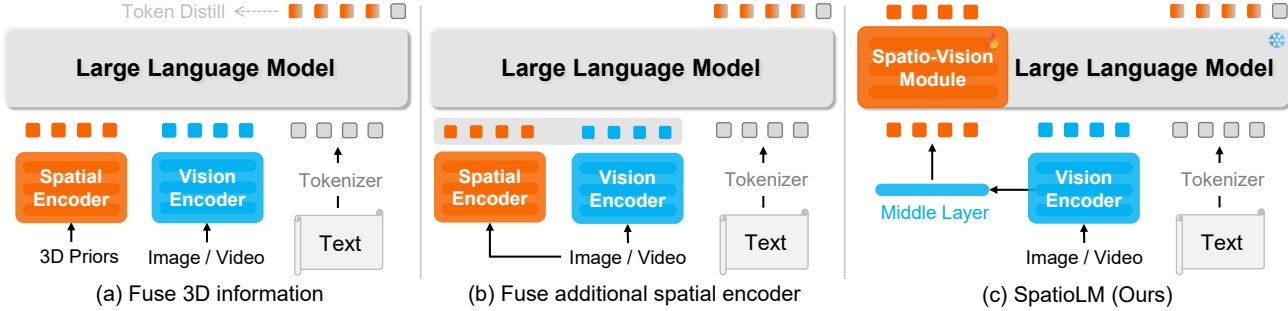

*Figure 2.* (a) Methods that explicitly leverage 3D priors, such as depth maps, point clouds, or camera parameters. (b) Methods that introduce an additional spatial encoder to inject 3D information. (c) Ours: a purely 2D vision-language framework that elicits 3D geometric structure from pretrained VLM visual tokens, without relying on explicit 3D inputs.

ing capabilities. A third prominent paradigm addresses 3D reasoning through tool-use and agentic frameworks, leveraging LLMs to execute code or call external APIs (e.g., depth estimators, 3D detectors) (Surís et al., 2023; Marsili et al., 2025; Gupta & Kembhavi, 2023), or utilizing reinforcement learning and verifiers without explicit 3D supervision (Sarch et al., 2026; Marsili & Gkioxari, 2025). While these methods offer significant advantages by circumventing invasive fine-tuning and providing high flexibility in zero-shot settings, they inherently treat the VLM as a high-level planner reliant on external tools. This introduces multi-step execution latency, cascading errors from API failures, and leaves the intrinsic spatial reasoning capabilities of the visual backbone fundamentally unimproved.

Overall, existing methods face a dilemma: they either rely on external dependencies (explicit 3D prior inputs or tool APIs) or embed spatial awareness through costly architectural changes that degrade general-purpose capabilities. This leaves open a critical question: how can the latent spatial structure in pretrained VLMs be elicited end-to-end on demand without sacrificing their general-purpose intelligence? This gap motivates **SpatioLM**, a unified purely 2D vision-language framework that incorporates a plug-and-play Spatio-Vision Module to enhance intrinsic spatial intelligence while preserving the general capabilities of pretrained VLMs.

## 3. Method

We propose **SpatioLM**, a unified framework that enhances VLMs with general physical spatial intelligence. As illustrated in Fig. 3, SpatioLM freezes a pretrained VLM backbone and introduces a parameter-efficient runtime *plug-and-play Spatio-Vision Module* (SV-Module, which consists of stacked Spatio-Vision Blocks) to elicit the geometric knowledge inherent in VLMs by extracting intermediate vision tokens. This parameter-efficient design enables a unified model to perform spatial perception and understanding, and can be adapted to downstream vision-language-action (VLA) tasks with minimal modification.

### 3.1. Problem definition and notation

The input sequence of the model is defined as the following tuple format:

$$X = \left( X^v, X^t \right), \tag{1}$$

where $X^v \in \mathbb{R}^{T \times H \times W \times 3}$ denotes a visual input sequence ($T = 1$ for images and $T > 1$ for videos), and $X^t$ is the corresponding textual instruction. The model generates a textual response $Y^t = (y_1, \ldots, y_L)$, where $L$ is the generated sequence length. During training, SpatioLM additionally predicts dense geometric outputs

$$Y^g = (\bar{D}, \bar{\mathcal{R}}), \tag{2}$$

where $\bar{D}$ and $\bar{\mathcal{R}}$ serve as the pseudo-labels for the depth and camera ray map, respectively. The model is formulated as an autoregressive conditional distribution, and its mathematical formulation is given by:

$$
\begin{aligned}
P(Y^t, Y^g \mid X^v, X^t) &= P(Y^g \mid X^v, X^t) \\
&\prod_{i=1}^{L} P(y_i \mid y_{<i}, Y^g, X^v, X^t).
\end{aligned} \tag{3}
$$

### 3.2. Spatio-Vision Module

To leverage the strong semantic capacity of VLMs while minimizing computational overhead, we adopt a parameter-efficient, ControlNet-inspired side-tuning strategy. Rather than focusing solely on parameter efficiency, this design aims to *efficiently elicit the spatial knowledge inherent in VLMs* while preserving their general-purpose capability.

**Visual Condition.** We employ a pretrained VLM constructed with a vision encoder and an LLM, in which all parameters of the vision encoder and the LLM remain frozen throughout the training process. Given the visual input $X^v$, the vision tokens generated by the vision encoder are mathematically denoted as:

$$\mathcal{H}_0^v = E_v(X^v), \quad \mathcal{H}_0^v \in \mathbb{R}^{T \times N \times C}, \tag{4}$$

where $\mathcal{H}_0^v$ are the vision tokens derived from the vision encoder. Inspired by recent findings (Meng et al., 2024; Bai

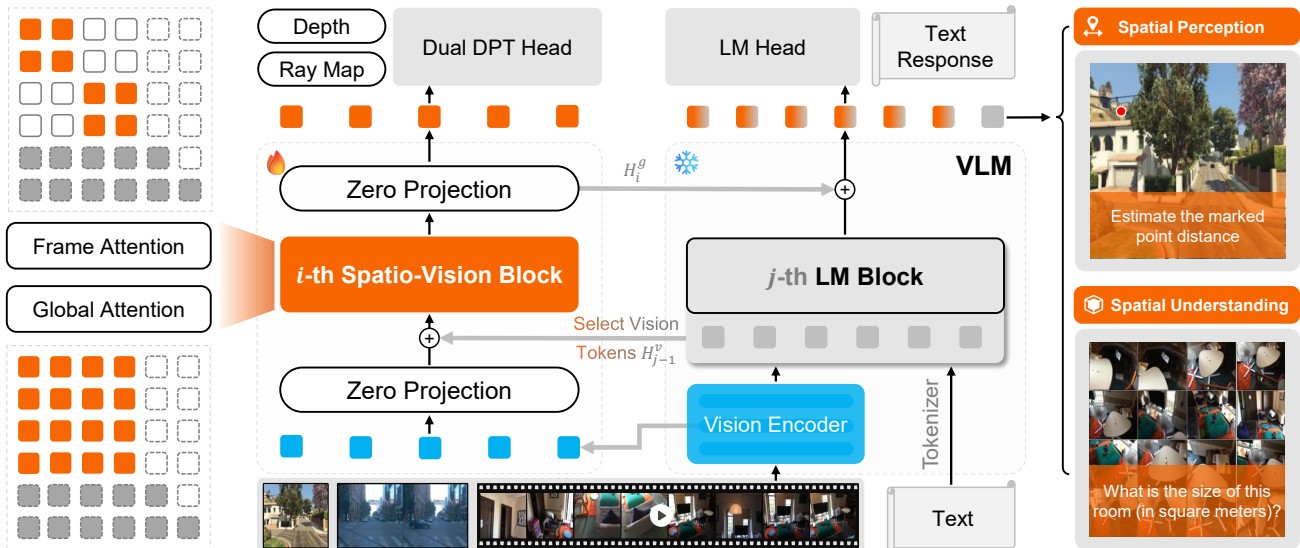

*Figure 3.* Overview of SpatioLM. SpatioLM augments a frozen VLM with a plug-and-play Spatio-Vision Module. The Spatio-Vision Module elicits geometry-aware features from visual tokens and injects them into language blocks via zero-initialized projections, enabling visual spatial reasoning while preserving VLM's general-purpose ability. Training incorporates auxiliary pseudo depth and camera supervision, while inference requires no additional 3D priors and is conducted through a unified text generation interface.

et al., 2025a; Bolya et al., 2026) that intermediate ViT layers capture richer spatial information for geometric tasks than the final layer, we utilize tokens from the intermediate layers as the initial inputs for our Spatio-Vision Block. Specifically, we select the 16th layer from the 24-layer ViT, guided by empirical analysis showing that layers around 16/24 yield the strongest spatial embeddings (Bolya et al., 2026). We denote these intermediate tokens as $\mathcal{H}_0^{v'}$ to distinguish them from the final-layer vision tokens $\mathcal{H}_0^v$ fed into the LLM. Here, $N$ and $C$ denote the number of tokens per frame and the feature dimension, respectively.

**Spatial Feature Elicitation.** To elicit spatial representations, in the $i$-th Spatio-Vision Block, we process the hidden features input to the $j$-th LM block, thereby extracting geometry-aware tokens $\mathcal{H}_i^g$. Subsequently, we fuse the extracted $\mathcal{H}_i^g$ with the output of the vision-language tokens by the $j$-th LM block, which then serves as input to the $(j+1)$-th LM Block. Note that we elicit the geometry-aware tokens of LM Blocks at regular intervals via Spatio-Vision Blocks, with a strict one-to-one mapping between each Spatio-Vision Block and a single LM Block. Hence, the index correspondence between Spatio-Vision Blocks and LM Blocks follows $j = k + i \times s$, where $k$ denotes the initial LM Block for the initiation of geometry-aware tokens elicitation, $i$ denotes the $i$-th Spatio-Vision Block, and $s$ denotes the number of LM Blocks between successive intervals. Let $\mathcal{H}_{j-1}^m \in \mathbb{R}^{M \times C}$ denote the multimodal tokens from $(j-1)$-th LM Block, where $M$ is the total length of vision-language tokens. $\mathcal{I}^v$ is the position index set of vision tokens in the multimodal sequence; the vision tokens from the $(j-1)$-th LM block are then denoted as $\mathcal{H}_{j-1}^v = \mathcal{H}_{j-1}^m[\mathcal{I}^v]$. Specif-

ically, $\mathcal{I}^v$ corresponds to the indices of all vision tokens at the current layer, which are clearly distinguished from linguistic token indices. Furthermore, the geometry-aware tokens elicited by the Spatio-Vision Block are denoted as:

$$\mathcal{H}_i^g = M_{sv}\left(\mathcal{H}_{j-1}^v + \mathcal{P}_z(\mathcal{H}_{i-1}^g)\right), \ \mathcal{H}_i^g \in \mathbb{R}^{|\mathcal{I}^v| \times C}, \quad (5)$$

where $M_{sv}$ denotes the computation of the Spatio-Vision Block and $\mathcal{P}_z$ represents the operation of the zero-initialized projection layer. In Eq. (5), $\mathcal{H}_{j-1}^v = \mathcal{H}_0^{v'}$ if $j-1=0$. To integrate the elicited geometry-aware tokens into the frozen language model without disrupting its pretrained distribution, we employ the zero-initialized projection layer to fuse the geometry-aware tokens with those of the LM Block, which is mathematically denoted as:

$$\hat{\mathcal{H}}_j^m[\mathcal{I}^v] = \mathcal{H}_j^m[\mathcal{I}^v] + \mathcal{P}_z(\mathcal{H}_i^g), \quad (6)$$

where $\hat{\mathcal{H}}_j^m$ denotes the vision-language token incorporating geometry-aware features, which replaces $\mathcal{H}_j^m$ in the pretrained language model. Subsequently, $\hat{\mathcal{H}}_j^m$ is processed within the standard frozen LM blocks.

**Spatio-Vision Block.** The Spatio-Vision Block is the core component responsible for explicit 3D perception. To effectively capture both intra-frame geometric structures and inter-frame geometric consistency, we adopt an *Alternating-Attention* mechanism inspired by VGGT (Wang et al., 2025a). This mechanism allows the model to separately model fine-grained geometry within each frame and global geometric relationships across frames. Specifically, each SV-Block alternates once between:

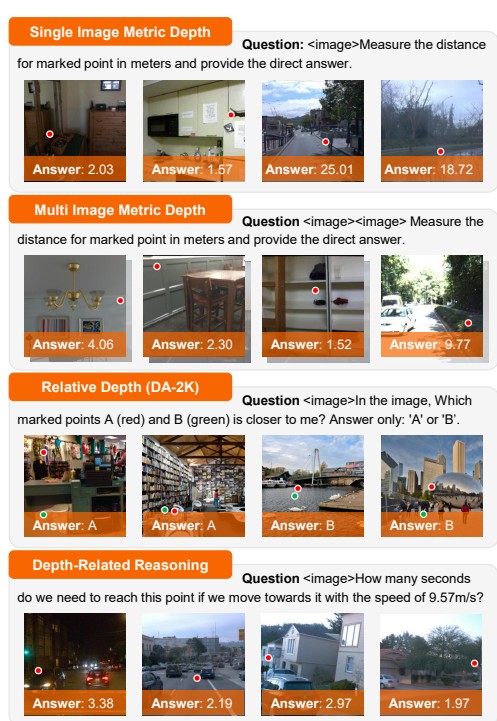

*Figure 4.* Qualitative depth-tasks on the MD-S, MD-M, DA-2K, and DR benchmarks.

*Table 1.* **Overall evaluation results on depth benchmarks.** Metric depth estimation for single-image (MD-S) and multi-images (MD-M) settings measured by $\delta < 1.25 \uparrow$, relative depth estimation on DA-2K, and depth-related multi-task benchmarks (DR); *doubao-1.5* is abbreviation for *doubao-1.5-thinking-vision-pro-250428*. Results marked with * are re-evaluated by us under the official settings, following the same setting throughout the paper. The best and runner-up results are **bolded** and underlined, respectively.

| Methods | MD-S | MD-M | DA-2K | DR |
|---|---|---|---|---|
| *Proprietary Models (API)* | | | | |
| GPT-5.2* (OpenAI, 2025) | 15.5 | 21.4 | 56.0 | 11.2 |
| Gemini-2.5-flash* (Comanici et al., 2025) | 28.6 | 28.7 | 75.5 | 9.87 |
| Qwen3-VL-plus* (Bai et al., 2025a) | 37.6 | 37.9 | 79.4 | 17.5 |
| Doubao-1.5* (Guo et al., 2025) | 52.6 | 41.1 | 80.7 | 13.6 |
| *Open-source Models* | | | | |
| LLaVA-NeXT-7B* (Zhang et al., 2024b) | 9.24 | 10.0 | 51.2 | 6.46 |
| Qwen2.5-VL-72B* (Bai et al., 2025b) | 32.9 | 31.5 | 64.0 | 12.4 |
| InternVL3.5-8B* (Wang et al., 2025b) | 41.0 | 28.3 | 60.3 | 10.2 |
| Qwen3-VL-235B-A22B* (Bai et al., 2025a) | 48.4 | 39.4 | 73.4 | 17.2 |
| *Specialized Spatial Models* | | | | |
| Cosmos-R1-7B* (Azzolini et al., 2025) | 19.4 | 25.4 | 53.1 | 7.01 |
| VST-7B-SFT* (Yang et al., 2025b) | 51.0 | 26.3 | **83.8** | 10.9 |
| Cambrian-S-7B* (Yang et al., 2025c) | 3.09 | 3.21 | 53.7 | 1.74 |
| SenseNovaSI-8B* (Cai et al., 2025b) | 52.6 | 37.7 | 71.8 | 17.5 |
| DepthLM(Pixtral-12B)* (Cai et al., 2025c) | 18.0 | 40.0 | - | 11.9 |
| *Ours* | | | | |
| SpatioLM$_{\text{InternVL3.5-8B}}$ | 50.9 | 32.0 | 81.7 | 24.1 |
| SpatioLM$_{\text{SenseNovaSI-8B}}$ | **83.5** | **69.0** | **83.8** | **41.8** |

- **Frame Attention**, which independently attends to visual tokens within each frame, preserving fine-grained geometric details.

- **Global Attention**, which jointly attends to all tokens across frames, enabling the aggregation of multi-view or temporal geometric consistency.

We utilize $K=6$ SV-Blocks by default and each SV-Block contains 50M parameters. Importantly, this alternating attention is applied *exclusively* inside the SV-Block and does not modify the causal self-attention mechanism of the frozen LLM decoder.

**Dual DPT Head.** To enforce explicit geometric learning, the output features $H^g$ are fed into a dual-branch Dense Prediction Transformer (DPT) head adopted from Depth Anything V3 (Lin et al., 2025). The two branches predict:

- **Dense Depth Map** $\hat{D}$, which may represent metric or relative depth depending on the supervision.

- **Camera Ray Map** $\hat{\mathcal{R}}$, which encodes per-pixel viewing directions from camera intrinsics and extrinsics.

By supervising these dense geometric predictions, the injected features are encouraged to encode camera-aware 3D priors that are crucial for downstream spatial tasks.

### 3.3. Loss Design

The overall training objective is a weighted sum of a language modeling loss and mixed geometric losses.

**Language Modeling Loss.** We employ the standard cross-entropy loss for text generation:

$$\mathcal{L}_L = -\sum_{i=1}^{L} \log P(y_i \mid y_{<i}, X^v, X^t). \quad (7)$$

**Distillation Loss.** To align the SV-Block features with robust 3D representations, we apply Vision Token Supervision (VTS) by distilling them toward the latent features of a pretrained 3D foundation model (Depth Anything V3 (Lin et al., 2025)). Specifically, given student visual features $H_s$ from the SV-Module and teacher features $H_t$ (spatially aligned and projected to the same dimension), we match their token-structure similarity via a Gram-matrix:

$$\mathcal{L}_d = \left\| H_s H_s^\top - H_t H_t^\top \right\|_F. \quad (8)$$

**Dense Geometric Loss.** The dual DPT head is supervised with dense depth and ray-map targets, which we refer to as Dense Geometric Supervision (DGS):

$$\mathcal{L}_g = \mathcal{L}_{de}(\hat{D}, \bar{D}) + \lambda \mathcal{L}_{ray}(\hat{\mathcal{R}}, \bar{\mathcal{R}}), \quad (9)$$

where $\bar{D}$ and $\bar{\mathcal{R}}$ are the pseudo-labels, and $\mathcal{L}_{de}$ and $\mathcal{L}_{ray}$ are the depth and ray losses designed for the Depth Anything V3 model (Lin et al., 2025), $\hat{D}$ and $\hat{\mathcal{R}}$ are the depth and camera parameter values predicted by the model.

**Overall Objective.** The final training loss is

$$\mathcal{L} = \alpha\mathcal{L}_L + \beta\mathcal{L}_d + \gamma\mathcal{L}_g, \quad (10)$$

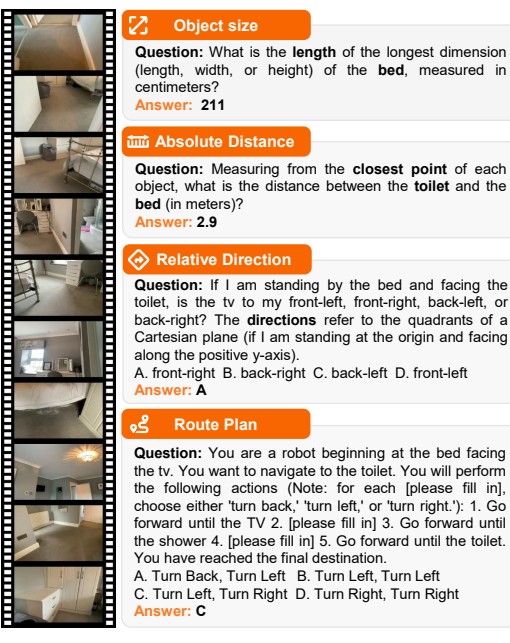

*Figure 5.* Qualitative tasks on VSI-Bench.

*Table 2.* **Evaluation results on VSI-bench.** The best and runner-up results are **bolded** and underlined, respectively.

| Methods | Avg. | Obj. Cht. | Abs. Dist. | Obj. Size | Room Size | Rel. Dist. | Rel. Dir. | Route Plan | Appr. Order |
|---|---|---|---|---|---|---|---|---|---|
| *Proprietary Models (API)* | | | | | | | | | |
| GPT-4o | 34.0 | 46.2 | 5.3 | 43.8 | 38.2 | 37.0 | 41.3 | 31.5 | 28.5 |
| Gemini-2.5 Pro | 51.5 | 43.8 | 34.9 | 64.3 | 42.8 | 61.1 | 47.8 | 45.9 | 71.3 |
| *Open-source Models* | | | | | | | | | |
| LLaVA-OneVision-72B | 40.2 | 43.5 | 23.9 | 57.6 | 37.5 | 42.5 | 39.9 | 32.5 | 44.6 |
| LLaVA-NeXT-Video-72B | 40.9 | 48.9 | 22.8 | 57.4 | 35.3 | 42.4 | 36.7 | 35.0 | 48.6 |
| InternVL3.5-8B* | 56.1 | 70.7 | 43.3 | 68.1 | 63.6 | 55.6 | 49.1 | 36.6 | 61.5 |
| Qwen2.5-VL-7B* | 32.7 | 34.5 | 19.4 | 47.6 | 40.8 | 32.8 | 24.5 | 32.5 | 29.4 |
| Qwen3-VL-235B-A22B* | 51.0 | 60.6 | 41.6 | 73.5 | 63.8 | 47.7 | 43.7 | 35.1 | 42.1 |
| *Specialized Spatial Models* | | | | | | | | | |
| Spatial-MLLM-4B | 48.4 | 65.3 | 34.8 | 63.1 | 45.1 | 41.3 | 46.2 | 33.5 | 46.3 |
| VLM-3R-7B | 60.9 | 70.2 | 49.4 | 69.2 | 67.1 | 65.4 | 80.5 | 45.4 | 40.1 |
| GS-Reasoner (w/d) | 64.7 | 69.1 | **61.9** | 70.0 | 65.7 | 65.4 | 88.9 | 44.3 | 52.3 |
| Cambrian-S-7B | 67.5 | 73.2 | 50.5 | 74.9 | **72.2** | 71.1 | 76.2 | 41.8 | 80.1 |
| VST-SFT-7B | 60.6 | 72.0 | 44.4 | 74.3 | 68.3 | 59.7 | 55.8 | 44.9 | 65.2 |
| SenseNovaSI-8B* | 68.7 | 75.7 | 53.2 | 72.8 | 67.5 | 77.0 | 72.7 | **48.5** | 82.6 |
| *Ours* | | | | | | | | | |
| SpatioLM$_{\text{InternVL3.5-8B}}$ | 65.0 | 76.1 | 50.9 | 73.6 | 65.1 | 73.3 | 70.1 | 44.3 | 66.9 |
| SpatioLM$_{\text{SenseNovaSI-8B}}$ | **71.6** | **81.9** | 55.7 | **75.2** | 71.5 | **78.2** | 74.4 | 47.9 | **87.8** |

where $\alpha$, $\beta$, and $\gamma$ weight the objectives and are set to 0.6, 0.2, and 0.2. During training, only SV-Module and projection layers are updated, with the VLM backbone frozen.

### 3.4. Unified Task Formulation

SpatioLM frames diverse spatial tasks as conditional language modeling, using prompt design to specify tasks and next-token prediction in the vocabulary space.

**Spatial Perception.** Spatial perception tasks in SpatioLM are depth-centric, with metric depth estimation as the core objective. Following DepthLM (Cai et al., 2025c), metric depth estimation is formulated as a question-answering task. Visual markers are overlaid on the input image, and the prompt asks for the distance at the marked location.

**Spatial Understanding.** Spatial understanding is formulated as a visual question answering, addressing spatial tasks including, but not limited to, object counting, relative size estimation, spatial relations, and route planning.

## 4. Experiments

### 4.1. Experimental Setup

**Training Data.** To systematically enhance the spatial intelligence of SpatioLM, we construct a comprehensive training corpus that unifies low-level spatial perception with high-level spatial understanding, spanning diverse indoor and outdoor scenes across both real-world and synthetic domains. To ensure rigorous evaluation and prevent data leakage, all training samples are strictly sourced from the official training splits of the respective datasets, reserving the

validation and test splits exclusively for evaluation. Further details of the data construction and task design are provided in Appendix B.1.

**Training Setting.** We consider two configurations:

- **SpatioLM$_{\text{SenseNovaSI-8B}}$**: InternVL3 architecture, initialized from SenseNovaSI-8B (Cai et al., 2025b).

- **SpatioLM$_{\text{InternVL3.5-8B}}$**: InternVL3.5 architecture, initialized from InternVL3.5-8B (Wang et al., 2025b).

All models are trained on 64 NVIDIA H200 GPUs using AdamW ($\beta_1 = 0.9$, $\beta_2 = 0.95$, weight decay = 0.1), with a cosine learning rate schedule and a warm-up ratio of 0.1. Training is conducted for 600 epochs on spatial perception tasks and 2 epochs on spatial understanding tasks.

**Evaluation Setting.** During evaluation, the Dual DPT Head can be removed. The SV-Module is forwarded once during prefilling to encode spatial context, after which token generation follows standard VLM decoding with key–value caching and incurs no additional inference overhead. This design enables enhanced physical spatial reasoning with efficient and scalable inference.

### 4.2. Visual Spatial Reasoning

#### 4.2.1. SPATIAL PERCEPTION

**Benchmarks.** To evaluate low-level spatial perception of our SpatioLM, we adopt four complementary benchmarks: metric depth estimation on single-image (MD-S) and multi-image (MD-M) settings, relative depth estimation on DA-2K (Yang et al., 2024), and depth-related multi-task

reasoning (DR). MD-S, MD-M, and DR are constructed by us. Details are provided in Appendix B.2.

**Baselines.** We compare against three groups of baselines: proprietary closed-source, open-source, and spatial-specialized models.

**Results.** Tab. 1 demonstrates that SpatioLM$_{SenseNovaSI-8B}$ achieves the best overall performance across all benchmarks (83.5 on MD-S, 69.0 on MD-M, 83.8 on DA-2K, and 41.8 on DR). It significantly outperforms strong baselines on both single-image and multi-image metric depth, attains the highest accuracy on DA-2K, and demonstrates more stable generalization on a range of depth-related reasoning tasks. Compared with the corresponding base models (SenseNovaSI-8B/InternVL3.5-8B), SpatioLM yields consistent and substantial improvements; it also remains competitive against the strongest proprietary, open-source, and specialized spatial models, highlighting its ability to better integrate visual cues, geometric relations, and metric reasoning for general spatial perception and physical reasoning. More detailed results are reported in Appendix C.

### 4.2.2. SPATIAL UNDERSTANDING

*Table 3.* **Evaluation results on ScanQA and SQA3D.** ScanQA is evaluated with BLEU-1 (B1), BLEU-4 (B4), METEOR (M), ROUGE-L (R), and CIDEr (C), while SQA3D uses exact match accuracy (E1) and its refined variant (ER1). The best and runner-up results are **bolded** and underlined, respectively.

| Methods | ScanQA$_{val}$ | | | | | SQA3D$_{test}$ | |
|---|---|---|---|---|---|---|---|
| | B1 | B4 | M | R | C | E1 | ER1 |
| *Task-Specific Models* | | | | | | | |
| ScanQA | 30.2 | 10.1 | 13.1 | 33.3 | 64.9 | 47.2 | – |
| SQA3D | 30.5 | 11.2 | 13.5 | 34.5 | – | 46.6 | – |
| *Open-source Models* | | | | | | | |
| Qwen2.5-VL-72B | 26.8 | 12.0 | 13.0 | 35.2 | 66.9 | 47.0 | 50.9 |
| InternVL3.5-8B* | 35.1 | 8.12 | 14.6 | 39.4 | 74.2 | 51.0 | 54.0 |
| Qwen3-VL-8B* | 14.1 | 8.34 | 11.6 | 35.8 | 63.0 | 47.2 | 51.4 |
| *Specialized Spatial Models* | | | | | | | |
| SenseNovaSI-8B* | 26.4 | 0.00 | 13.4 | 37.6 | 70.0 | 47.8 | 50.1 |
| VST-7B-SFT* | 8.43 | 0.50 | 9.63 | 41.9 | 82.1 | 45.7 | 48.7 |
| Spatial-MLLM-4B | 44.4 | 14.8 | 18.4 | 45.0 | 91.8 | 55.9 | **58.7** |
| *Ours* | | | | | | | |
| SpatioLM$_{InternVL3.5-8B}$ | 45.4 | 16.2 | 19.0 | 47.5 | 98.2 | 55.3 | 57.7 |
| SpatioLM$_{SenseNovaSI-8B}$ | **48.4** | **16.3** | **19.8** | **48.9** | **101.8** | **56.5** | 58.6 |

**Benchmarks.** To evaluate high-level spatial understanding of SpatioLM, we adopt three representative 3D spatial reasoning benchmarks: VSI-Bench (Yang et al., 2025a), ScanQA (val) (Azuma et al., 2022), and SQA3D (test) (Ma et al., 2022). Details are provided in Appendix B.2.

**Baselines.** We evaluate SpatioLM against diverse baselines: VSI-Bench includes proprietary, open-source and specialized spatial models, while ScanQA and SQA3D include task-specific, open-source and specialized models.

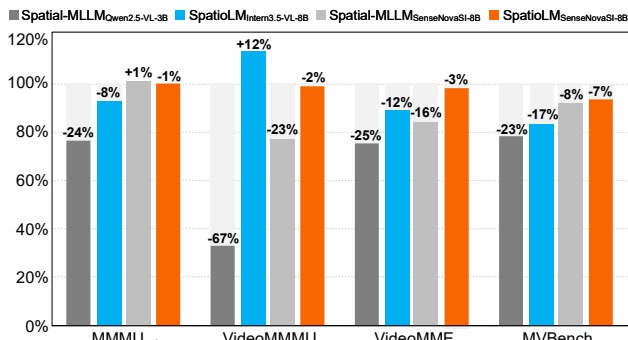

*Figure 6.* Performance drop on general-purpose capabilities.

**Results.** We present the quantitative results on Tab. 2 and Tab. 3; SpatioLM$_{SenseNovaSI-8B}$ achieves the best performance across all three benchmarks. On VSI-Bench, it ranks $1^{st}$ with Avg = 71.6, the first to exceed 70, and outperforms all proprietary, open-source and specialized spatial models. On ScanQA, our approach consistently outperforms all baselines across all metrics, indicating enhanced understanding of spatial relations and object identification in 3D scenes. On SQA3D, it achieves 56.5 / 58.6 in terms of E1 / ER1, surpassing SenseNovaSI-8B (47.8 / 50.1) by a clear margin, demonstrating stronger spatial reasoning under pose-conditioned question answering settings. More detailed results are reported in Appendix C.

### 4.3. Evaluation on General Capabilities

To assess whether our approach preserves general-purpose capabilities of VLMs, we evaluate SpatioLM on representative general benchmarks, including VideoMME (Fu et al., 2025), VideoMMMU (Hu et al., 2025), MVBench (Li et al., 2024), and MMMU (val) (Yue et al., 2024). We compare against Spatial-MLLM (Wu et al., 2025), a representative method that enhances spatial intelligence via an external spatial encoder; the results are summarized in Fig. 6. Results indicate that, despite large-scale training targeted at spatial intelligence, the performance degradation of SpatioLM on general tasks is considerably smaller than that of baseline models. Its overall performance remains on par with the baseline and even improves on certain benchmarks (e.g., +12% on VideoMMMU). In contrast, Spatial-MLLM exhibits a substantial decline in general benchmarks, with reductions ranging from 23% to 67%. These findings demonstrate that SpatioLM enhances spatial intelligence while effectively mitigating catastrophic forgetting, thereby preserving general-purpose capabilities. More quantitative details are provided in Tab. 14 of Appendix C.

### 4.4. Downstream Task on Embodied Manipulation

To evaluate the capabilities of SpatioLM in physical spatial intelligence, we adapt SpatioLM-2B (built upon SenseNovaSI-2B (Cai et al., 2025b) following the same

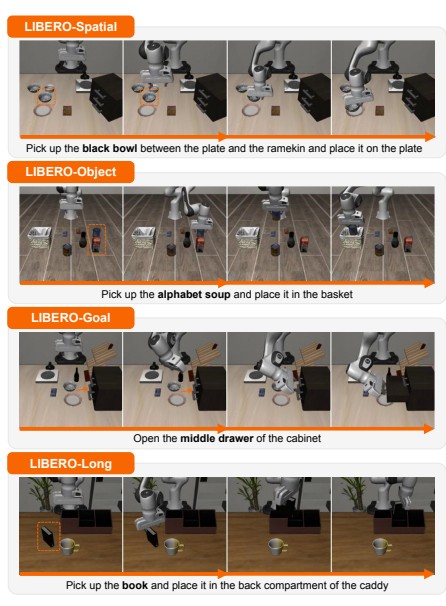

*Figure 7.* Qualitative manipulation-tasks on the LIBERO benchmark.

*Table 4.* **Evaluation results on LIBERO benchmark across 4 suites.** P.T: pretraining on large-scale robot manipulation data; A.T: action type (D: discrete, C: continuous). SenseNovaSI-VLA0 and SenseNovaSI-OFT denote the baseline models under the discrete-action and continuous-action settings, respectively. We train one policy for all 4 suites. All scores are averaged over 500 trials for each task suite (10 tasks × 50 episodes).

| Methods | P.T | A.T | Spatial | Object | Goal | Long | Avg. |
|---|---|---|---|---|---|---|---|
| OpenVLA (Kim et al., 2024) | ✓ | D | 84.7 | 88.4 | 79.2 | 53.7 | 76.5 |
| SpatialVLA (Qu et al., 2025) | ✓ | D | 88.2 | 88.9 | 78.6 | 55.5 | 78.1 |
| $\pi_0$-FAST (Black et al., 2024) | ✓ | D | 96.4 | 96.8 | 88.6 | 60.2 | 85.5 |
| FlowVLA (Zhong et al., 2025) | ✗ | D | 93.2 | 95.0 | 91.6 | 72.6 | 88.1 |
| UniVLA (Bu et al., 2025) | ✓ | D | 96.5 | 96.8 | 95.6 | 92.0 | 95.2 |
| SmolVLA (Shukor et al., 2025) | ✗ | C | 93.0 | 94.0 | 91.0 | 77.0 | 88.8 |
| OpenVLA-OFT (Kim et al., 2025) | ✗ | C | 94.3 | 95.2 | 91.7 | 86.5 | 91.9 |
| GR00T-N1 (Bjorck et al., 2025) | ✓ | C | 94.4 | 97.6 | 93.0 | 90.6 | 93.9 |
| $\pi_0$ (Black et al., 2024) | ✓ | C | **96.8** | 98.8 | 95.8 | 85.2 | 94.2 |
| DepthVLA (Yuan et al., 2025) | ✗ | C | 96.4 | 98.0 | 95.8 | 89.2 | 94.9 |
| SenseNovaSI-VLA0 | ✗ | D | 93.8 | 84.0 | 79.4 | 59.4 | 79.2 |
| SenseNovaSI-OFT | ✗ | C | 92.2 | 98.6 | 96.4 | 93.8 | 95.3 |
| SpatioLM-VLA0 | ✗ | D | 95.6 | 97.2 | 92.8 | 78.6 | 91.0 |
| SpatioLM-OFT | ✗ | C | 93.4 | **99.0** | **97.8** | **94.8** | **96.3** |

Spatio-Vision Module design) into a vision-language-action (VLA) model and instantiate two variants under different action modeling paradigms: *discrete action* and *continuous action*. The base baselines, SenseNovaSI-VLA0 and SenseNovaSI-OFT, denote SenseNovaSI-2B fine-tuned with the same VLA-0 and OpenVLA-OFT recipes, respectively.

- **SpatioLM-VLA0**: Under the discrete action setting, we follow VLA-0 (Goyal et al., 2025) paradigm and formulate embodied manipulation as a discretized action sequence prediction problem.
- **SpatioLM-OFT**: Under the continuous action setting, we follow OpenVLA-OFT (Kim et al., 2025) paradigm and directly regress continuous actions.

**Inference Strategy and Results.** At inference, we apply an action ensemble over a temporal sliding window. As shown in Tab. 4, SpatioLM improves manipulation success on LIBERO (Liu et al., 2023a): from 79.2% to 91.0% over SenseNovaSI-VLA0 in the discrete setting, and from 95.3% to 96.3% over SenseNovaSI-OFT in the continuous setting. Notably, SpatioLM-OFT achieves competitive performance among VLA methods with only 2B parameters, without large-scale robot manipulation pretraining, demonstrating strong parameter efficiency and spatial generalization.

### 4.5. Ablation Study

We conduct an ablation study on SpatioLM$_{SenseNovaSI-8B}$ to analyze the contributions of the Spatio-Vision Module (SV-Module), Vision Token Supervision (VTS), and Dense Geometric Supervision (DGS) (Tab. 5). Moreover, additional analyses examine the sensitivity to loss weights and the effect of the attention mechanism in the SV-Block.

**Effectiveness of SV-Module.** Integrating the SV-Module yields substantial gains across settings, improving MD-S by +22.1 (52.6→74.7) and DA-2K by +12.2 (71.8→84.0), while also delivering consistent improvements on VSI-Bench, validating the effectiveness of SV-Module for eliciting latent spatial knowledge.

*Table 5.* Ablation study on the SV-Module, VTS, and DGS.

| SV-Module | VTS | DGS | MD-S | DA-2K | VSI-Bench |
|---|---|---|---|---|---|
| ✗ | ✗ | ✗ | 52.6 | 71.8 | 68.7 |
| ✓ | ✗ | ✗ | 74.7 | 84.0 | 69.1 |
| ✓ | ✓ | ✗ | 76.9 | 82.4 | 69.6 |
| ✓ | ✗ | ✓ | 81.5 | 83.3 | 70.8 |
| ✓ | ✓ | ✓ | 83.5 | 83.8 | 71.6 |

**Effectiveness of VTS and DGS.** On top of the SV-Module, VTS and DGS provide complementary supervision: VTS enhances semantic consistency at the vision-token level, while DGS enforces fine-grained geometric constraints. Their combination achieves the best overall performance (83.5 on MD-S, 83.8 on DA-2K, 71.6 on VSI-Bench), indicating orthogonal and synergistic contributions that jointly fully exploit the capacity of the SV-Module.

**Sensitivity to Loss Weights.** The default loss weights $(\alpha, \beta, \gamma) = (0.6, 0.2, 0.2)$ were chosen empirically following common practice in geometry-supervised VLMs (Huang et al., 2025; Zheng et al., 2025). To evaluate robustness to this choice, we vary the weights and report results in Tab. 6. SpatioLM is not sensitive to these hyperparameters: VSI-Bench changes within only 0.5 points across all configurations, while ScanQA and SQA3D remain consistently strong. This indicates that the proposed supervision is stable

across a reasonable range of objective balances.

Table 6. Sensitivity analysis on loss weights.

| $\alpha$ | $\beta$ | $\gamma$ | VSI-Bench | ScanQA(C) | SQA3D(ER1) |
|---|---|---|---|---|---|
| 0.4 | 0.2 | 0.2 | 71.5 | 100.2 | 58.8 |
| 0.6 | 0.2 | 0.2 | 71.6 | 101.8 | 58.6 |
| 0.8 | 0.2 | 0.2 | 71.9 | 101.2 | 59.9 |
| 0.6 | 0.3 | 0.2 | 71.4 | 101.7 | 59.2 |
| 0.6 | 0.1 | 0.2 | 71.8 | 103.2 | 60.2 |
| 0.6 | 0.2 | 0.1 | 71.4 | 100.9 | 60.1 |
| 0.6 | 0.2 | 0.3 | 71.7 | 102.2 | 58.5 |
| 0.8 | 0.2 | 0.3 | 71.9 | 102.9 | 59.4 |

**Attention Mechanism.** We compare the proposed Alternating-Attention with a self-attention baseline in the SV-Block (Tab. 7). Alternating-Attention consistently outperforms self-attention, improving VSI-Bench by +1.2, ScanQA CIDEr by +3.1, and SQA3D ER1 by +1.8. These gains validate the benefit of explicitly decoupling intra-frame and inter-frame attention for capturing geometric consistency.

Table 7. Ablation on attention mechanism.

| Mechanism | VSI-Bench | ScanQA(C) | SQA3D(ER1) |
|---|---|---|---|
| Self-Attention | 70.4 | 98.7 | 56.8 |
| Alt.-Attention (Ours) | 71.6 | 101.8 | 58.6 |

**Choice of the Number of SV-Blocks.** We further study the choice of the number of SV-Blocks in the SV-Module (Tab. 8). Increasing the number of SV-Blocks from 4 to 6 yields clear performance gains, especially on MD-S (76.5→83.5) and DA-2K (82.4→83.8), indicating that sufficient stacking is necessary to fully capture spatial interactions. Further increasing the number of SV-Blocks to 8 or 12 does not bring additional benefits and leads to marginal performance saturation. Performance on VSI-Bench remains stable across different settings, demonstrating the robustness of the SV-Module. Based on this performance–efficiency trade-off, we adopt 6 SV-Blocks as the default setting, which achieves the best overall performance while requiring fine-tuning of only approximately 0.3B parameters, highlighting the parameter-efficient nature of our method.

Table 8. Ablation study on the number of SV-Blocks.

| SV-Blocks | MD-S | DA-2K | VSI-Bench |
|---|---|---|---|
| 4 | 76.5 | 82.4 | 71.3 |
| 6 | 83.5 | 83.8 | 71.6 |
| 8 | 83.0 | 83.1 | 71.7 |
| 12 | 82.9 | 83.7 | 71.5 |

## 5. Conclusion

We propose SpatioLM, a parameter-efficient framework integrated with a plug-and-play and non-invasive Spatio-Vision Module to address the core challenge that existing VLMs encounter increased complexity and degraded general-purpose capabilities. This module enhances spatial reasoning from low-level perception to high-level understanding without relying on explicit 3D priors or external spatial encoders. The non-invasive form of the module ensures improved visual spatial reasoning while preserving the model's general-purpose capabilities. Extensive experimental results demonstrate that SpatioLM outperforms state-of-the-art models on spatial tasks, effectively mitigates the degradation of general-purpose capabilities, and exhibits competitive transferability in downstream embodied manipulation tasks. We further note that spatially-aware foundation models and tool-use agentic systems are synergistic: a spatially-aware foundation model serves as a more reliable backbone for agentic systems, while tool-use frameworks maximize reasoning ceilings for complex tasks. Overall, this work offers a novel perspective for physical spatial intelligence exploration and an effective solution for subsequent research.

**Limitations.** Despite the effectiveness of SpatioLM, several limitations remain. (1) Pseudo-label dependence: training relies heavily on pseudo-labels generated by teacher models (e.g., Depth Anything V3). The inherent noise in these labels imposes an upper bound on the model's overall 3D comprehension, as errors inevitably propagate to the learned representations. (2) Motion-dynamics trade-off: as revealed by the performance drops in our MVBench metrics (Fig. 6), geometric supervision emphasizes static spatial structure, which partially competes with temporal-dynamic cues. This leads to degradation on motion-intensive subtasks (e.g., action counting, moving direction), while scene-level reasoning is preserved. Supplementing geometric supervision with explicit temporal signals is a promising future direction. (3) Safety-critical deployment: given the aforementioned constraints, SpatioLM may be unreliable for safety-critical applications. We strongly caution against deploying it in such scenarios without rigorous additional verification.

## Impact Statement

Addressing the critical limitation of multimodal large models in visual spatial reasoning, this work proposes SpatioLM, a parameter-efficient spatial VLMs that enhances spatial intelligence without relying on additional 3D priors or third-party spatial encoders. By significantly boosting VLMs' spatial understanding and geometric reasoning capabilities while retaining their general knowledge, our research advances the state-of-the-art in multimodal large models for spatial reasoning tasks. This progress not only facilitates further exploratory research on multimodal learning but also enables impactful downstream applications in embodied AI and autonomous driving domains, contributing to the responsible development and deployment of AI technologies that benefit society.

## Acknowledgments

We sincerely thank the reviewers, area chairs, and program chairs for their constructive feedback and valuable suggestions. This paper uses several public datasets and benchmarks for academic research, including ARKitScenes (license), ScanNet (terms of use), ScanNet++ (terms of use), Virtual KITTI 2 (terms of use), Waymo Open Dataset (license), and ScanQA (license), whose licenses or terms of use may restrict commercial use. The authors confirm that the use of these datasets and the model weights trained and/or evaluated with them in this paper is solely for academic research purposes and has not been used for any commercial activity.

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

# A. Related Work

## A.1. Spatial Perception

Spatial perception constitutes a fundamental component of spatial intelligence, aiming to recover metric scale and geometric structure from visual observations. Recent efforts attempt to extend Vision-Language Models (VLMs) beyond 2D semantic understanding toward metric depth estimation, narrowing the gap between visual semantics and physical geometry. A representative example is DepthLM (Cai et al., 2025c), which demonstrates that VLMs can achieve pixel-level metric depth estimation comparable to advanced pure vision models such as DepthPro (Bochkovskii et al., 2024) and Metric3Dv2 (Hu et al., 2024). This performance is largely enabled by intrinsic-conditioned preprocessing, most notably unifying camera focal length to 1000 pixels during training and inference. While effective under controlled settings, such reliance on camera-intrinsic normalization fundamentally limits applicability in open-world, RGB-only scenarios, where a calibrated imaging process cannot be assumed and geometric cues must be recovered implicitly from visual tokens. Related works such as SpatialVLM (Chen et al., 2024) and SpatialRGPT (Cheng et al., 2024) inject geometry indirectly by converting outputs of vision models into textual prompts for large language model reasoning. While conceptually lightweight, these pipeline-based designs are prone to error accumulation and typically emphasize object-level reasoning, making direct comparison with perception-oriented visual models difficult. SpatialBot (Cai et al., 2025a) instead relies on an external depth estimation module to enable pixel-level 3D understanding, but exhibits degraded performance on multi-view or multi-image tasks such as pose estimation. Collectively, these methods enhance depth awareness through auxiliary components, yet externalize geometric reasoning rather than grounding it intrinsically within the VLM, resulting in limited generalization and robustness on complex geometric scenarios.

## A.2. Spatial Understanding

Beyond metric depth, spatial understanding requires higher-level reasoning over object relations, layouts, and scene geometry. Existing methods largely follow three distinct paradigms: explicit geometric augmentation, architecture-level modification, and agentic tool-use frameworks.

The first paradigm enriches inputs with explicit 3D priors, either from sensors or auxiliary estimation models. Sensor-based approaches, typically relying on RGB-D input, demonstrate strong performance in controlled environments. OmniEVA (Liu et al., 2025a) dynamically fuses depth maps and camera parameters via task-adaptive 3D grounding and embodiment-aware reasoning, enabling question answering, localization, and navigation. Similarly, 3DRS (Huang et al., 2025) distills knowledge from a pretrained 3D foundation model (Wang et al., 2025a) to supervise 3D-aware representations in MLLMs, substantially improving multi-view correspondence and spatial tasks on VSI-Bench (Yang et al., 2025a). RoboRefer (Zhou et al., 2025) introduces a dedicated depth encoder to reason over 31 spatial relations in RGB-D scenes, achieving a high success rate on RefSpatial-Bench (Zhou et al., 2025), while MM-Spatial (Daxberger et al., 2025) explores depth-map and multi-view fusion for object counting and localization. Estimation-based methods remove direct sensor dependence but still rely on explicitly reconstructed geometry. GS-Reasoner (Chen et al., 2025b) combines semantic and geometric representations through autoregressive 3D grounding, achieving performance competitive with prior methods on VSI-Bench (Yang et al., 2025a), and establishing a new state of the art on Scan2Cap (Chen et al., 2021). SSR (Liu et al., 2025b) distills depth maps into structured rationales, improving spatial reasoning efficiency via knowledge distillation. While effective, both categories hinge on explicit 3D priors that are external to the VLM and often unavailable in unconstrained, open-world RGB settings, thereby limiting scalability and robustness.

The second paradigm introduces architectural modifications to embed spatial reasoning capacity more deeply into VLMs. SpatialLLM (Ma et al., 2025), VLM-3R (Fan et al., 2025), VG-LLM (Zheng et al., 2025), and Spatial-MLLM (Wu et al., 2025) augment pretrained models with specialized spatial encoders, extracting 3D cues from image or video and improving performance on spatial benchmarks. However, by introducing external spatial encoders, these approaches typically adopt a dual-encoder architecture, which necessitates careful feature alignment between the newly introduced spatial representations and the pretrained VLM feature space. To achieve effective integration, this alignment often requires fine-tuning the LLM or vision backbone on 3D-specific data. Such fine-tuning can disrupt the pretrained feature distribution and is prone to catastrophic forgetting of general semantic understanding and reasoning capabilities. In parallel, dataset-centric efforts (Chen et al., 2024; Yang et al., 2025b;c; Cai et al., 2025b; Hao et al., 2025) attempt to inject spatial knowledge through large-scale supervision, but remain confined to end-to-end VLM retraining. As a result, these approaches do not fundamentally address how the latent spatial structure within pretrained models can be selectively activated, rather than relearned.

The third paradigm addresses 3D reasoning through tool-use and agentic frameworks, leveraging LLMs to call external

APIs (e.g., depth estimators, 3D detectors) (Surís et al., 2023; Marsili et al., 2025; Gupta & Kembhavi, 2023), or utilizing RL-based verifiers without explicit 3D supervision (Sarch et al., 2026; Marsili & Gkioxari, 2025). ViperGPT (Surís et al., 2023) and VISPROG (Gupta & Kembhavi, 2023) demonstrate that language models can compose visual API calls via code generation to solve complex reasoning tasks without direct geometric training. VADAR (Marsili et al., 2025) further extends this paradigm to spatial reasoning with a dynamic API that orchestrates depth estimators and 3D detectors on demand. On the reinforcement-learning side, ViGoRL (Sarch et al., 2026) adopts grounded RL for visual reasoning, while Marsili and Gkioxari (Marsili & Gkioxari, 2025) train visual reasoners with multimodal verifiers, both improving spatial reasoning accuracy without explicit 3D supervision. While these agentic methods achieve impressive reasoning ceilings on specific tasks, they depend on the availability and reliability of external tools at inference time, and their spatial reasoning capability is not internalized within the model's representations. Notably, a spatially-aware foundation model and tool-use agents are complementary: a stronger spatial backbone serves as a highly reliable engine for agentic systems, while tool-use frameworks can further extend reasoning capabilities for highly complex tasks.

### A.3. Summary and Motivation

Across a broad range of spatial tasks, existing approaches have achieved substantial progress but remain fragmented in both design and capability. Prior methods face a fundamental dilemma: they either rely on external dependencies (such as explicit 3D prior inputs, auxiliary models, or dynamic tool APIs), which limit robustness in unconstrained settings, or they attempt to enhance intrinsic spatial reasoning ability through invasive architectural modifications and extensive fine-tuning that compromise pretrained general capabilities. These limitations directly motivate the central question of this work: how can latent spatial structure in pretrained VLMs be activated end-to-end on demand, without sacrificing their general-purpose intelligence? This gap motivates **SpatioLM**, which pursues a unified, non-intrusive, plug-and-play framework for enhancing intrinsic spatial perception and understanding capabilities while gracefully preserving the general-purpose capabilities of pretrained VLMs.

## B. Dataset Setup and Benchmarks

### B.1. Training Data

To systematically enhance the spatial intelligence of SpatioLM, we construct a comprehensive training corpus that unifies low-level spatial perception with high-level spatial understanding, spanning diverse indoor and outdoor scenes across both real-world and synthetic domains.

*Table 9.* Statistics of datasets used for spatial perception training. #Scenes (Repeated) denotes the number of scene-level samples after repeated scene sampling.

| Dataset | #Scenes (Repeated) | Environment | Data Type |
|---|---|---|---|
| ARKitScenes (Baruch et al., 2021) | 8,900 | Indoor | Real |
| Hypersim (Roberts et al., 2021) | 2,000 | Indoor | Synthetic |
| MVS-Synth (Huang et al., 2018) | 1,000 | Outdoor | Synthetic |
| ScanNet (Dai et al., 2017) | 2,000 | Indoor | Real |
| ScanNet++ (Yeshwanth et al., 2023) | 1,000 | Indoor | Real |
| Virtual-KITTI-2 (Cabon et al., 2020) | 1,000 | Outdoor | Synthetic |
| Waymo (Sun et al., 2020) | 5,000 | Outdoor | Real |
| Total | 20,900 | – | – |

**Spatial Perception Data.** For low-level spatial perception, we construct a diverse set of depth-centric spatial tasks, which can be grouped into four categories according to their supervision sources and design principles: (1) metric depth estimation on single-image; (2) metric depth estimation on multi-images; (3) relative depth estimation; (4) depth-related multi-tasks.

Following DepthLM (Cai et al., 2025c), we adopt the task formulations of single-image metric depth estimation and depth-related multi-tasks. The latter includes a set of geometric reasoning objectives, such as cross-view distance estimation between two points, speed estimation, time estimation, metric distance estimation between two points, and camera pose estimation. In addition, we extend metric depth estimation to the multi-images setting, where multi-images sampled from the same scene are jointly provided. This task explicitly encourages the model to integrate cross-view geometric cues

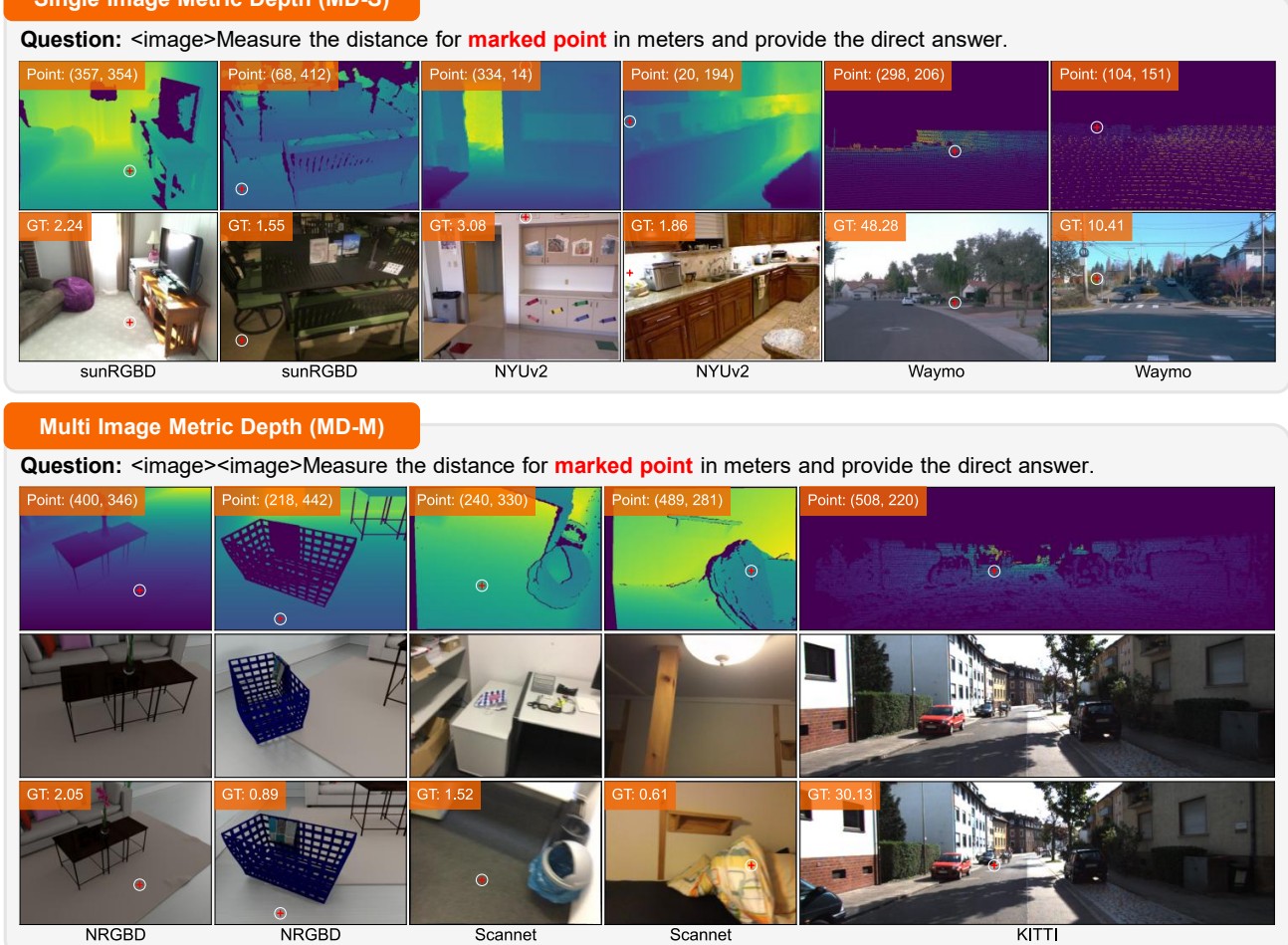

*Figure 8.* Visualization samples of our spatial perception benchmarks. The first row shows ground-truth single-image metric depth (MD-S), and the second row shows multi-image metric depth estimation (MD-M).

and reason about consistent metric depth across viewpoints, constituting an important extension of prior depth-centric supervision. In addition, relative depth estimation is constructed by following the task formulation of the public benchmark DA-2K (Yang et al., 2024), which provides a standardized and well-established protocol for relative depth reasoning. These tasks share a common reliance on underlying depth perception and collectively encourage robust reasoning over metric scale, 3D geometry, and motion. In contrast to DepthLM (Cai et al., 2025c), which relies on camera intrinsic normalization or specialized preprocessing, our approach directly builds supervision from raw RGB images without requiring camera intrinsic information, making it more aligned with real-world deployment scenarios.

The training data are aggregated from seven large-scale 3D datasets, including ScanNet (Dai et al., 2017), ScanNet++ (Yeshwanth et al., 2023), ARKitScenes (Baruch et al., 2021), Hypersim (Roberts et al., 2021), Waymo (Sun et al., 2020), MVS-Synth (Huang et al., 2018), and Virtual-KITTI-2 (Cabon et al., 2020). Together, these datasets span indoor and outdoor environments, static and dynamic scenes, and both real-world and photorealistic synthetic captures, providing broad coverage of scene geometry, appearance, and motion.

Each dataset consists of multiple scene segments; during training, one or two images are randomly sampled from a single scene, followed by random cropping and resizing to a resolution of $448 \times 448$. A visual marker is then randomly selected within the image to form a corresponding question-answer sample. Each training instance is represented as a quadruple $D_i = \langle I_i, M_i, Q_i, A_i \rangle$, denoting the input image, visual marker, question, and answer, respectively.

To mitigate data source bias and improve cross-domain generalization, we apply repeated scene-level sampling (scene statistics reported in Tab. 9) and online sampling during training, balancing geometric complexity, lighting variation, and

viewpoint diversity. This strategy provides a robust foundation for recovering metric scale and 3D geometry from monocular inputs.

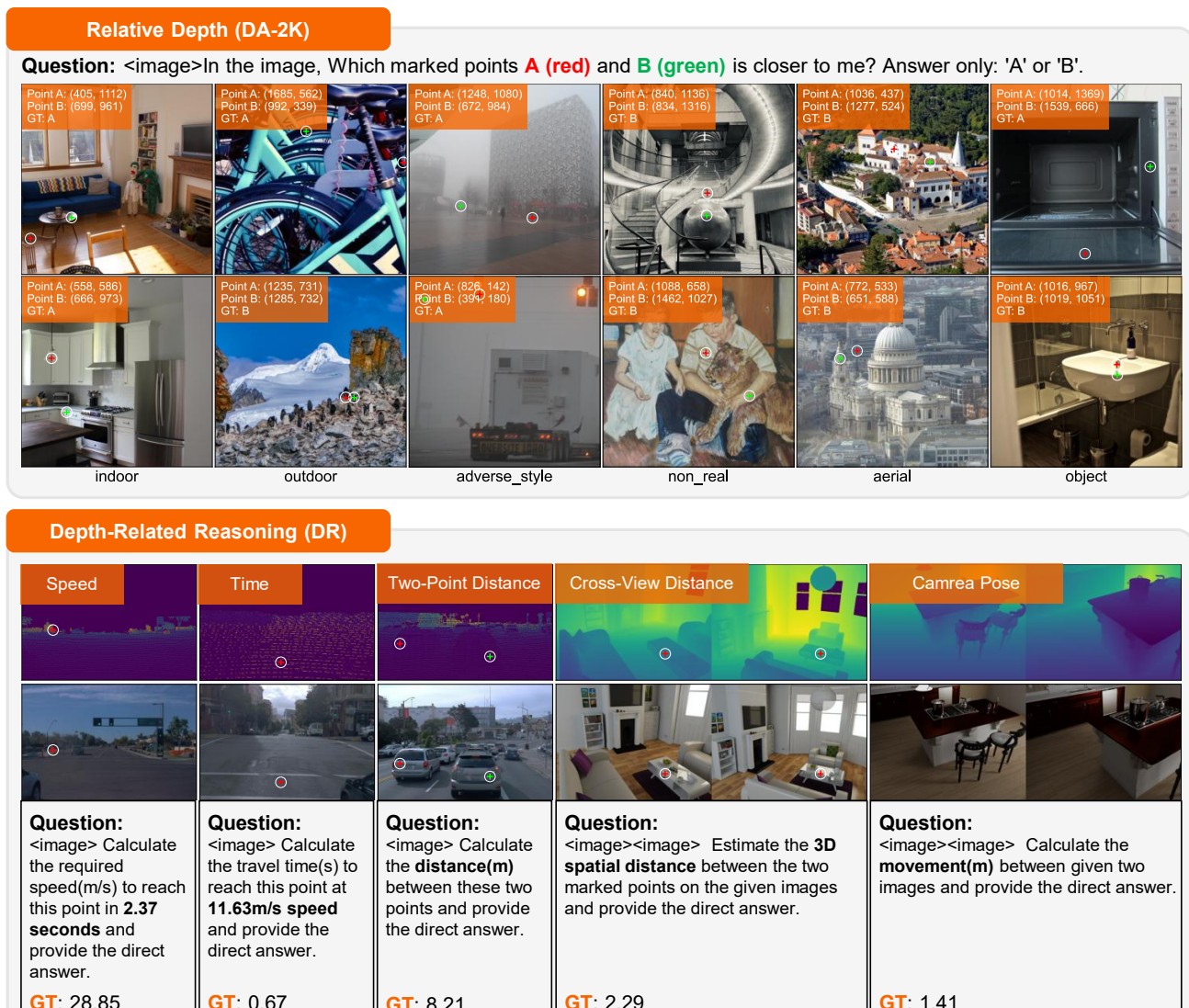

*Figure 9.* Visualization samples of spatial perception benchmarks. The first row shows relative depth estimation (DA-2K), and the second row shows depth-related multi-tasks (DR).

**Spatial Understanding Data.** For high-level spatial understanding, we leverage large-scale 3D visual question answering (VQA) datasets that require comprehensive spatial reasoning over complex scenes, including VSI-590K (Yang et al., 2025c), ScanQA (Azuma et al., 2022), and SQA3D (Ma et al., 2022).

VSI-590K (Yang et al., 2025c) is a large-scale instruction-tuning dataset comprising 590K spatially grounded instruction–response pairs over images and videos, collected from 10 distinct sources. It supports a wide range of visual spatial reasoning tasks, including object counting, relative distance and direction estimation, route planning, object and room size estimation, absolute distance estimation, and appearance order reasoning, thereby offering diverse and comprehensive spatial supervision.

ScanQA (Azuma et al., 2022) is built on 800 ScanNet (Dai et al., 2017) scenes and provides visual question–answer pairs that focus on spatial relations and semantic reasoning in real-world 3D indoor environments, grounding language understanding in accurate scene geometry. SQA3D (Ma et al., 2022) further extends ScanQA (Azuma et al., 2022) by introducing more diverse reasoning questions, covering spatial relation comprehension, commonsense reasoning, navigation, and multi-hop

reasoning, thus encouraging deeper spatial and semantic understanding.

For consistency, image data are resized to $448 \times 448$, while video data are uniformly sampled into 32 frames before applying the same resolution normalization. In total, the spatial understanding corpus comprises over 643K training samples.

Together with the depth-centric spatial perception tasks, these high-level 3D VQA datasets enable SpatioLM to learn visual spatial reasoning in a hierarchical manner, bridging low-level metric geometry with high-level semantic and relational understanding.

*Table 10.* Detailed evaluation results on MD-S and MD-M benchmarks. The best and runner-up results are **bolded** and underlined, respectively.

| Methods | MD-S | | | | | | | | MD-M | | | | | | | |
|---|---|---|---|---|---|---|---|---|---|---|---|---|---|---|---|---|
| | Avg. | | SUN RGB-D | | NYU-v2 | | Waymo | | Avg. | | NRGBD | | ScanNet | | KITTI | |
| | $\delta\uparrow$ | A.R$\downarrow$ | $\delta\uparrow$ | A.R$\downarrow$ | $\delta\uparrow$ | A.R$\downarrow$ | $\delta\uparrow$ | A.R$\downarrow$ | $\delta\uparrow$ | A.R$\downarrow$ | $\delta\uparrow$ | A.R$\downarrow$ | $\delta\uparrow$ | A.R$\downarrow$ | $\delta\uparrow$ | A.R$\downarrow$ |
| *Proprietary Models (API)* | | | | | | | | | | | | | | | | |
| GPT-5.2 | 15.5 | 0.697 | 18.4 | 0.595 | 13.2 | 0.647 | 14.8 | 0.848 | 21.4 | 0.658 | 28.6 | 0.433 | 21.5 | 0.603 | 14.0 | 0.938 |
| Gemini-2.5-flash | 28.6 | 0.523 | 42.8 | 0.354 | 36.0 | 0.367 | 7.00 | 0.839 | 28.7 | 0.485 | 36.0 | 0.399 | 35.4 | 0.390 | 14.8 | 0.667 |
| Qwen3-vl-plus | 37.6 | 0.391 | 58.8 | 0.189 | 49.6 | 0.229 | 4.33 | 0.755 | 37.9 | 0.303 | 27.0 | 0.329 | 61.4 | 0.213 | 25.2 | 0.367 |
| Doubao-1.5-thinking-vision | 52.6 | 0.430 | 61.2 | 0.226 | 79.2 | 0.164 | 17.5 | 0.900 | 41.1 | 0.417 | **53.8** | **0.270** | 63.6 | 0.239 | 6.00 | 0.741 |
| *Open-source Models* | | | | | | | | | | | | | | | | |
| LLaVA-NeXT-Video-7B | 9.24 | 9.61 | 5.80 | 1.20 | 6.60 | 0.705 | 15.3 | 26.9 | 9.99 | 44.6 | 6.80 | 1.150 | 9.76 | 1.36 | 13.4 | 131 |
| Qwen2.5-VL-72B | 32.9 | 1.18 | 37.2 | 0.445 | 43.6 | 0.428 | 18.0 | 2.68 | 31.5 | 1.26 | 41.2 | 0.428 | 27.2 | 0.670 | 26.2 | 2.67 |
| InternVL3.5-8B | 41.0 | 0.794 | 59.8 | 0.211 | 54.8 | 0.208 | 8.50 | 1.96 | 28.3 | 0.751 | 30.2 | 0.336 | 48.6 | 0.292 | 6.20 | 1.63 |
| Qwen3-VL-235B-A22B | 48.4 | 0.379 | 67.2 | 0.182 | 61.4 | 0.212 | 16.5 | 0.742 | 39.9 | 0.385 | 29.8 | 0.328 | 58.9 | 0.246 | 31.0 | 0.582 |
| *Specialized Spatial Models* | | | | | | | | | | | | | | | | |
| Cosmos-R1-7B | 19.4 | 29.2 | 29.8 | 0.706 | 22.2 | 0.741 | 6.33 | 386 | 25.4 | 115 | 26.6 | 0.415 | 29.5 | 0.900 | 20.0 | 344 |
| VST-7B-SFT | 51.0 | 6.18 | 76.8 | 0.147 | 66.4 | 0.186 | 9.67 | 18.2 | 26.3 | 0.419 | 24.4 | 0.374 | 46.3 | 0.251 | 8.20 | 0.632 |
| Cambrian-S-7B | 3.09 | 0.780 | 5.60 | 0.745 | 1.00 | 0.745 | 2.67 | 0.849 | 3.21 | 0.711 | 3.00 | 0.631 | 2.22 | 0.715 | 4.40 | 0.786 |
| SenseNovaSI-8B | 52.6 | 0.431 | 77.2 | 0.178 | 76.0 | 0.181 | 4.67 | 0.934 | 37.7 | 0.407 | 40.4 | 0.282 | 71.3 | 0.198 | 1.40 | 0.742 |
| DepthLM(Pixtral-12B) | 18.0 | 0.745 | 8.60 | 0.593 | 28.6 | 0.426 | 16.7 | 1.22 | 40.0 | 0.472 | 29.4 | 0.437 | 42.9 | 0.359 | 47.8 | 0.619 |
| *Ours* | | | | | | | | | | | | | | | | |
| SpatioLM$_{\text{InternVL3.5-8B}}$ | 50.9 | 4.09 | 77.0 | 0.141 | 65.6 | 0.189 | 10.0 | 11.9 | 32.0 | 3.78 | 33.6 | 0.360 | 57.1 | 0.208 | 5.40 | 10.8 |
| SpatioLM$_{\text{SenseNovaSI-8B}}$ | **83.5** | **0.310** | **88.5** | **0.136** | **84.6** | **0.128** | **77.5** | **0.666** | **69.0** | **0.216** | 47.6 | 0.360 | **84.1** | **0.127** | **75.2** | **0.161** |

## B.2. Evaluation Benchmarks

**Spatial Perception Benchmarks.** To evaluate the low-level spatial perception capabilities of SpatioLM, we adopt four depth-centric spatial benchmarks: metric depth estimation on single images (MD-S) and multiple images (MD-M), relative depth estimation on DA-2K (Yang et al., 2024), and depth-related multi-tasks (DR). Among them, MD-S, MD-M, and DR are constructed by us, while DA-2K is a public benchmark adopted for standardized evaluation.

**MD-S.** The single-image depth estimation benchmark follows standard evaluation protocols and is constructed using SUN RGB-D (Song et al., 2015), NYUv2 (Silberman et al., 2012), and Waymo (Sun et al., 2020), with 500 evaluation samples per dataset. For each RGB image, we randomly select 10 spatial locations and retrieve the corresponding metric depth values from the aligned depth map to form metric depth question-answer pairs. Depth is predicted without access to camera intrinsics or extrinsics, enabling a fair evaluation of monocular depth recovery. Multiple visualization examples of the benchmark construction are shown in the first row of Fig. 8.

**MD-M.** The multi-image depth estimation benchmark evaluates the model's ability to reason over cross-view geometric cues. It is constructed using NRGBD (Azinović et al., 2022), ScanNet (Dai et al., 2017), and KITTI (Geiger et al., 2013), with 500 evaluation samples per dataset. Similar to MD-S, depth values are queried at randomly selected image locations without relying on camera intrinsic or extrinsic information, enabling consistent evaluation of multi-view depth recovery. Multiple visualization examples of the benchmark construction are shown in the second row of Fig. 8.

**DA-2K.** For relative depth estimation, we adopt DA-2K (Yang et al., 2024), a public benchmark introduced in Depth Anything V2. DA-2K consists of 1K images with 2K pairwise relative depth annotations, covering eight diverse scenarios, including indoor, outdoor, non-realistic, transparent/reflective, adverse style, aerial, underwater, and object-centric scenes.

Multiple visualization examples of the benchmark are shown in the first row of Fig. 9. During evaluation, we shuffle the answer choices and report the average accuracy to ensure a fair and reliable comparison.

**DR.** Depth-related multi-tasks includes speed estimation, time estimation, two-point distance measurement, cross-view reasoning, and camera pose estimation. These tasks are constructed using data from Waymo (Sun et al., 2020) and 7-Scenes (Shotton et al., 2013), and jointly assess the model's ability to reason about metric scale, 3D geometry, and motion. Multiple visualization examples of the benchmark construction are shown in the second row of Fig. 9.

**Spatial Understanding Benchmarks.** To evaluate the high-level spatial understanding capabilities of SpatioLM, we adopt three representative 3D spatial reasoning benchmarks: VSI-Bench (Yang et al., 2025a), ScanQA (val) (Azuma et al., 2022), and SQA3D (test) (Ma et al., 2022).

**VSI-Bench.** VSI-Bench is a video-based visual-spatial intelligence benchmark of over 5,000 question-answer pairs. It covers a broad range of spatial reasoning tasks, including configurational reasoning *(object count, relative distance, relative direction, route plan)*, measurement estimation *(object size, room size, and absolute distance)*, and spatiotemporal reasoning *(appearance order)*, under both numeric free-form and multiple-choice settings.

**ScanQA.** ScanQA is a benchmark for 3D question answering that evaluates spatial relation understanding and object identification in 3D scenes, grounding language queries in accurate scene geometry. For ScanQA, we evaluate performance using BLEU-1 (B1), BLEU-4 (B4), METEOR (M), ROUGE-L (R), and CIDEr (C).

**SQA3D.** SQA3D investigates spatial understanding by combining situation understanding with situated reasoning, focusing on pose-conditioned question answering given the viewer's position and orientation. Since SQA3D contains deterministic answers, we evaluate performance using exact match accuracy (E1) and its refined variant (ER1).

*Table 11.* Detailed evaluation results on DA-2K benchmark. We shuffle the answer choices and report the average accuracy. The best and runner-up results are **bolded** and underlined, respectively.

| Methods | Avg. | Adv./Sty. | Aerial | Indoor | Non-real | Object | Outdoor | Tra./Ref. | Und./Wat. |
|---|---|---|---|---|---|---|---|---|---|
| *Proprietary Models (API)* | | | | | | | | | |
| GPT-5.2 | 56.0 | 64.0 | 53.6 | 56.4 | 60.4 | 56.8 | 50.0 | 48.1 | 55.6 |
| Gemini-2.5-flash | 75.5 | 75.3 | 67.0 | 76.7 | 79.5 | 76.4 | 72.7 | 72.0 | 88.9 |
| Qwen3-vl-plus | 79.4 | 81.7 | 62.9 | 78.1 | **92.7** | 80.4 | 77.3 | 74.8 | 83.8 |
| Doubao-1.5-thinking-vision-pro | 80.7 | 81.7 | **78.9** | 75.0 | 89.4 | 83.1 | 81.7 | 72.4 | 87.2 |
| *Open-source Models* | | | | | | | | | |
| LLaVA-NeXT-Video-7B | 51.2 | 50.9 | 48.5 | 52.4 | 49.5 | 48.6 | 54.7 | 53.3 | 46.2 |
| Qwen2.5-VL-72B | 64.0 | 65.2 | 56.7 | 63.1 | 71.9 | 66.9 | 60.8 | 57.5 | 72.6 |
| InternVL3.5-8B | 60.3 | 61.3 | 58.8 | 59.0 | 60.7 | 62.2 | 54.9 | 61.2 | 76.1 |
| Qwen3-VL-235B-A22B | 73.4 | 73.2 | 65.5 | 72.6 | 89.1 | 73.0 | 66.6 | 68.7 | 78.6 |
| *Specialized Spatial Models* | | | | | | | | | |
| Cosmos-R1-7B | 53.1 | 50.6 | 48.5 | 52.4 | 52.5 | 48.6 | 57.6 | 63.1 | 46.2 |
| VST-7B-SFT | **83.8** | 83.8 | **78.9** | 83.3 | 89.8 | 78.4 | **82.6** | 83.6 | **89.7** |
| Cambrian-S-7B | 53.7 | 54.3 | 54.6 | 53.3 | 53.8 | 55.4 | 54.9 | 53.3 | 46.2 |
| SenseNovaSI-2B | 59.0 | 62.8 | 56.2 | 56.9 | 60.7 | 64.2 | 55.2 | 54.2 | 70.1 |
| SenseNovaSI-8B | 71.8 | 72.6 | 66.0 | 65.0 | 78.5 | 73.6 | 68.9 | 73.8 | 88.9 |
| *Ours* | | | | | | | | | |
| SpatioLM$_{\text{InternVL3.5-8B}}$ | 81.7 | 82.9 | 76.3 | 82.1 | 88.8 | 81.1 | 77.6 | 77.1 | 88.9 |
| SpatioLM$_{\text{SenseNovaSI-8B}}$ | **83.8** | **84.5** | 73.2 | **85.2** | 88.8 | **85.1** | 77.9 | **87.9** | **89.7** |

# C. Additional Experiments

## C.1. Comparison with LoRA

To further compare SpatioLM with parameter-efficient fine-tuning of the frozen backbone, we evaluate a LoRA baseline with rank $= 16$ and $\alpha = 64$, using a parameter budget comparable to our SV-Module. As shown in Tab. 12, SpatioLM outperforms LoRA across all three spatial understanding benchmarks, improving VSI-Bench by $+1.8$, ScanQA CIDEr by $+4.6$, and SQA3D ER1 by $+2.9$. These results suggest that the proposed non-invasive side-path design is more effective for spatial reasoning than directly adapting the backbone with parameter-efficient fine-tuning.

*Table 12.* Comparison with LoRA under a comparable parameter budget. LoRA is applied with rank $= 16$ and $\alpha = 64$.

| Method | VSI-Bench | ScanQA(C) | SQA3D(ER1) |
|---|---|---|---|
| LoRA (rank $= 16$, $\alpha = 64$) | 69.8 | 97.2 | 55.7 |
| SpatioLM (Ours) | **71.6** | **101.8** | **58.6** |

## C.2. Detailed Evaluation Results on Metric Depth Benchmarks

We report detailed evaluation results for all metric depth benchmarks in Tab. 10, Tab. 11, and Tab. 13.

*Table 13.* Detailed evaluation results on DR benchmark. The best and runner-up results are **bolded** and underlined, respectively.

| Methods | Avg. | Speed | Time | Two-Point Distance | Cross-View Distance | Camera Pose |
|---|---|---|---|---|---|---|
| *Proprietary Models (API)* | | | | | | |
| GPT-5.2 | 11.2 | 4.33 | 8.33 | 4.17 | 33.8 | 5.20 |
| Gemini-2.5-flash | 9.87 | 2.50 | 10.5 | 6.33 | 18.8 | 11.2 |
| Qwen3-vl-plus | 17.5 | 15.3 | 0.00 | 12.5 | **34.4** | 21.4 |
| Doubao-1.5-thinking-vision-pro | 13.6 | 22.7 | 14.0 | 14.0 | 11.6 | 5.60 |
| *Open-source Models* | | | | | | |
| LLaVA-NeXT-Video-7B | 6.46 | 1.83 | 5.17 | 7.50 | 14.4 | 3.40 |
| Qwen2.5-VL-72B | 12.4 | 24.7 | 8.00 | 11.3 | 11.2 | 7.00 |
| InternVL3.5-8B | 10.2 | 10.5 | 4.83 | 10.5 | 22.6 | 2.60 |
| Qwen3-VL-235B-A22B | 17.2 | 20.3 | 3.50 | 14.0 | **34.4** | 13.6 |
| *Specialized Spatial Models* | | | | | | |
| Cosmos-R1-7B | 7.01 | 18.0 | 7.67 | 3.00 | 4.20 | 2.20 |
| VST-7B-SFT | 10.9 | 1.33 | 2.17 | 5.83 | 29.2 | 16.2 |
| Cambrian-S-7B | 1.74 | 2.83 | 0.37 | 0.00 | 2.12 | 3.40 |
| SenseNovaSI-8B | 17.5 | 7.17 | 11.17 | 5.67 | 32.6 | **30.8** |
| DepthLM(Pixtral-12B) | 11.9 | 19.5 | 3.83 | **17.5** | 13.8 | 4.80 |
| *Ours* | | | | | | |
| SpatioLM$_{\text{InternVL3.5-8B}}$ | 24.1 | 44.7 | 31.8 | 9.00 | 8.20 | 26.6 |
| SpatioLM$_{\text{SenseNovaSI-8B}}$ | **41.8** | **78.2** | **71.7** | 14.2 | 16.8 | 28.2 |

As shown in Tab. 10, on both MD-S and MD-M benchmarks' $\delta$, SpatioLM$_{\text{SenseNovaSI-8B}}$ achieves the best performance among proprietary closed-source models, open-source models, and specialized spatial models. Moreover, our spatially enhanced models consistently outperform their corresponding base models. Specifically, on MD-S, SpatioLM$_{\text{SenseNovaSI-8B}}$ improves over SenseNovaSI-8B from 52.6 to 83.5, and SpatioLM$_{\text{InternVL3.5-8B}}$ improves over InternVL3.5-8B from 41.0 to 50.9. Similar gains are observed on MD-M, where SpatioLM$_{\text{SenseNovaSI-8B}}$ improves from 37.7 to 69.0 and SpatioLM$_{\text{InternVL3.5-8B}}$ improves from 28.3 to 32.0, demonstrating the effectiveness of our spatial modeling across single- and multi-image metric depth estimation.

As shown in Tab. 11, on the DA-2K benchmark, our model SpatioLM$_{\text{SenseNovaSI-8B}}$ achieves highly competitive performance across proprietary closed-source, open-source, and specialized spatial models. Moreover, our models consistently outperform their corresponding base models. Specifically, SpatioLM$_{\text{SenseNovaSI-8B}}$ improves over SenseNovaSI-8B from 71.8 to 83.8, and SpatioLM$_{\text{InternVL3.5-8B}}$ improves over InternVL3.5-8B from 60.3 to 81.7, demonstrating the effectiveness of our spatial modeling for relative depth estimation.

As shown in Tab. 13, on the DR benchmark, our models SpatioLM$_{\text{SenseNovaSI-8B}}$ and SpatioLM$_{\text{InternVL3.5-8B}}$ achieve the first and second best results, with scores of 41.8 and 24.1, respectively. Compared to their corresponding base models, SpatioLM$_{\text{SenseNovaSI-8B}}$ improves from 17.5 to 41.8, while SpatioLM$_{\text{InternVL3.5-8B}}$ improves from 10.2 to 24.1, showing substantial gains. These results indicate that our spatial modeling effectively enhances spatial reasoning performance across diverse depth-related tasks, particularly in speed and time estimation.

Furthermore, as observed from the results in Tab. 10 and Tab. 13, there is a significant performance divergence across different models. Notably, Cambrian-S-7B exhibits near-zero performance (e.g., 3.09 and 3.21 in Tab. 10, and 1.74 in Tab. 13). One might intuitively attribute such performance gaps to a lack of close-domain training data. However, we clarify that this divergence primarily stems from intrinsic differences in task formulation and model design, rather than an unfair data advantage.

Specifically, while Cambrian-S-7B demonstrates strong capabilities in broad spatial reasoning tasks, its suboptimal performance on our specific benchmarks is attributed to task granularity misalignment. Given that both models share extensive training scenes (e.g., ScanNet, Hypersim) in their training corpora, Cambrian-S-7B was optimized for multiple-choice relative relations rather than point-level exact absolute depth regression.

To further disentangle training data from model design, we highlight two facts. First, SpatioLM achieves strong zero-shot generalization on strictly unseen domains (e.g., SUN RGB-D, NYU-v2), proving robust generalization over in-domain overfitting. Second, under an identical QA-based task formulation, SpatioLM still significantly outperforms equivalent baselines (e.g., DepthLM), firmly validating our architectural superiority over mere data scale or domain coverage.

### C.3. Detailed Evaluation Results on General-purpose Capabilities

Tab. 14 presents the performance of the proposed model (SpatioLM) and other comparative models relative to the base model across multiple tasks, including VideoMME, MVBench, MMMU, and VideoMMMU. As illustrated in Table 14, the average performance degradation of the two SpatioLM variants compared to the base model is maintained within 4% across various general tasks. In sharp contrast, the Spatial-MLLM model exhibits an average performance drop exceeding 40% relative to the base model on these general tasks. This clearly demonstrates that SpatioLM can retain excellent general capabilities when built on different base models. Furthermore, in certain specific tasks such as the adaptation task on the VideoMMMU dataset, SpatioLM achieves certain performance improvements compared to the baseline model. This finding further corroborates SpatioLM's significant advantage in effectively preserving general-purpose capabilities while boosting spatial understanding and geometric perception.

*Table 14.* Performance on general-purpose capabilities benchmarks: VideoMME, MVBench, MMMU (val), and VideoMMMU. Avg. denotes the average performance over all benchmarks for each model.

| Methods | Avg. | VideoMME | MVBench | MMMU (val) | VideoMMMU | | |
|---|---|---|---|---|---|---|---|
| | | | | | perception | comprehension | adaptation |
| BaseModel$_{\text{InternVL3.5-8B}}$ | 52.8 | 65.2 | 73.2 | 56.3 | 35.7 | 50.7 | 35.7 |
| SpatioLM$_{\text{InternVL3.5-8B}}$ | 50.5($-4\%$) | 57.7 | 60.5 | 51.8 | 35.0 | 42.0 | 55.7 |
| BaseModel$_{\text{SenseNovaSI-8B}}$ | 49.8 | 62.5 | 65.6 | 50.6 | 53.0 | 36.3 | 30.7 |
| SpatioLM$_{\text{SenseNovaSI-8B}}$ | 48.1($-3\%$) | 60.7 | 60.7 | 50.1 | 51.7 | 33.7 | 31.7 |
| BaseModel$_{\text{Qwen2.5-VL-3B}}$ | 49.8 | 58.5 | 67.0 | 47.2 | 56.3 | 39.7 | 30.3 |
| Spatial-MLLM$_{\text{Qwen2.5-VL-3B}}$ | 27.9($-44\%$) | 43.6 | 51.9 | 35.7 | 10.0 | 7.33 | 18.7 |

### C.4. Failure Cases and MVBench Degradation Analysis

**Failure Case Analysis.** We further analyze representative failure patterns on VSI-Bench. As shown in Tab. 15, shared failures concentrate on route planning, where SpatioLM and the base model fail on highly overlapping samples. This task requires long-horizon memory and continuous updates of spatial reference frames, indicating a limitation inherited from the base VLM. In contrast, the overlap is much lower on relative direction, where SpatioLM substantially reduces the number of failures, suggesting that the SV-Module effectively improves local spatial relation reasoning.

*Table 15.* Failure consistency analysis on VSI-Bench. Consistency denotes the ratio of shared failures over SpatioLM failures.

| Task Type | Ours Fail | Base Fail | Shared | Consistency |
|---|---|---|---|---|
| Route Planning | 101 | 100 | 96 | 95.0% |
| Rel. Direction | 120 | 166 | 64 | 53.3% |

**MVBench Degradation Analysis.** We decompose MVBench into 20 subtasks and compare SpatioLM against the corresponding frozen base model on both backbones. The performance degradation is concentrated in a small set of motion/dynamics-related tasks (Tab. 16), while scene-level reasoning tasks are largely preserved or even improved. This pattern indicates that geometry-oriented supervision emphasizes static spatial structure, which can compete with fine-grained temporal cues such as motion direction, dynamic attributes, and temporal counting. In contrast, holistic scene reasoning aligns better with 3D spatial understanding. This motivates future work that augments the SV-Module with explicit temporal-dynamic supervision, such as optical flow or motion-aware objectives.

*Table 16.* Detailed MVBench subtask results. Δ denotes the score change from the base model to SpatioLM on the same backbone.

| Category | Subtask | InternVL3.5-8B | | | SenseNovaSI-8B | | |
|---|---|---|---|---|---|---|---|
| | | Base | SpatioLM | Δ | Base | SpatioLM | Δ |
| Motion Dyn. | Moving Direction | 80.5 | 50.0 | -30.5 | 65.0 | 40.0 | -25.0 |
| | Moving Attribute | 97.0 | 71.5 | -25.5 | 92.0 | 81.5 | -10.5 |
| | Moving Count | 79.5 | 55.5 | -24.0 | 69.0 | 65.5 | -3.5 |
| Action Sem. | Action Antonym | 88.5 | 60.5 | -28.0 | 63.5 | 35.5 | -28.0 |
| | Action Count | 70.5 | 35.0 | -35.5 | 46.0 | 51.0 | +5.0 |
| | Action Sequence | 83.0 | 73.0 | -10.0 | 73.5 | 71.0 | -2.5 |
| | Action Localization | 67.5 | 54.0 | -13.5 | 64.0 | 60.5 | -3.5 |
| | Fine-Grained Action | 42.5 | 39.0 | -3.5 | 41.0 | 37.5 | -3.5 |
| Object/State | Object Existence | 97.0 | 79.0 | -18.0 | 90.5 | 83.5 | -7.0 |
| | State Change | 72.5 | 53.0 | -19.5 | 59.5 | 55.0 | -4.5 |
| | Character Order | 77.0 | 62.0 | -15.0 | 77.0 | 72.0 | -5.0 |
| | Fine-Grained Pose | 82.5 | 64.5 | -18.0 | 75.5 | 76.5 | +1.0 |
| | Counterfactual Infer. | 78.5 | 64.0 | -14.5 | 67.5 | 69.0 | +1.5 |
| Scene Reason. | Scene Transition | 89.5 | 88.0 | -1.5 | 89.5 | 89.5 | 0.0 |
| | Episodic Reasoning | 56.5 | 59.5 | +3.0 | 47.0 | 47.5 | +0.5 |
| | Unexpected Action | 77.0 | 74.5 | -2.5 | 75.0 | 72.5 | -2.5 |
| | Action Prediction | 61.0 | 57.5 | -3.5 | 58.5 | 54.5 | -4.0 |
| | Object Interaction | 82.5 | 73.5 | -9.0 | 80.5 | 78.0 | -2.5 |
| | Object Shuffle | 43.5 | 34.5 | -9.0 | 40.0 | 41.0 | +1.0 |
| | Egocentric Navigation | 36.5 | 38.0 | +1.5 | 36.5 | 31.5 | -5.0 |

## C.5. Qualitative Results of Spatial Perception

To visually demonstrate the performance of the SpatioLM model on spatial perception tasks, we present the visualization results of representative tasks, including MD-S, MD-M, DA-2K, and DR, with the corresponding effects illustrated in Fig. 10 and Fig. 11. In particular, Fig. 10 visualizes the performance of the SpatioLM model on the MD-S and DA-2K tasks, furnishing qualitative empirical evidence for the model's spatial reasoning capabilities. For the MD-S task, the color-coded depth maps generated by SpatioLM exhibit a high degree of consistency with the geometric structures of scenes. Furthermore, the errors between the estimated distances of marked points and the ground-truth values of samples fall within a controllable range, which verifies that the model can convert visual features into accurate depth values across a diverse range of indoor and outdoor scenes. In the DA-2K relative depth task, the depth maps produced by SpatioLM consistently reflect the spatial hierarchical relationships among all marked points with high accuracy. For instance, in the living room scene, the closer marked Point A (sofa) presents a warmer color tone than the distant marked Point B (fireplace), which validates the model's robust inferential capability for relative distances across diverse scene settings. The aforementioned visualization results demonstrate that SpatioLM achieves effective depth value estimation and relative distance perception across various indoor and outdoor single-image scenarios.

In addition, Fig. 11 presents the visualization results of the SpatioLM model on the Multi-Image Metric Depth (MD-M) task, encompassing input scene images, quantitative depth estimates for marked points, and color-coded depth maps, which validate the model's spatial perception performance from both qualitative and quantitative perspectives. This study employs representative indoor scenes (e.g., living rooms, kitchens, and storage rooms) from the NRGBD and ScanNet datasets, as well as outdoor street scenes from the KITTI dataset, as test samples to verify the model's cross-scenario generalization capability. Quantitatively, the model achieves a depth estimation error of less than 0.3 meters in close-range indoor scenes (e.g., ScanNet kitchens and storage rooms); even in long-range outdoor scenarios such as KITTI streets, the depth estimates remain within a reasonable range, demonstrating the model's robust inferential capability across diverse spatial scales. Qualitatively, the color-coded depth maps generated by SpatioLM exhibit a clear spatial hierarchical structure, where the tonal differentiation between foreground and background regions is highly consistent with scene geometric characteristics. This confirms that the model can convert multi-image visual cues into continuous and consistent depth representations. Overall, the SpatioLM model is capable of generating highly spatially consistent depth maps and producing plausible quantitative depth estimates in the MD-M task.

As shown in Fig. 12, we further visualize SpatioLM on DR Bench to assess its generalization on depth-related reasoning tasks. The first two rows show the inputs. For speed estimation, time estimation, and two-point distance measurement, the

model takes a single image, whereas camera pose estimation uses an image pair for cross-view comparison. The third row reports the model's generated textual reasoning and final predictions. The fourth and fifth rows show the depth map and the corresponding camera ray map predicted by our model's Dual DPT Head, which reveal the geometric cues and viewing-ray constraints exploited during inference. Qualitatively, in road scenes, the predicted depth exhibits stable ordinal structure across cars, the road surface, and distant buildings, with coherent boundaries (e.g., vehicle contours and road–sidewalk separations). Meanwhile, the ray maps follow the camera imaging geometry and provide an explicit line-of-sight constraint for lifting 2D observations to 3D space. This joint representation of depth distribution and ray geometry enables the model to extract reliable scale and relative pose signals, which are then aligned with the language reasoning process for tasks requiring metric distance estimation. Overall, these visualizations suggest that SpatioLM not only produces plausible spatial reasoning in language, but also leverages intermediate geometric representations (depth and camera rays) that are consistent with the final textual outputs, supporting our claim of general physical spatial intelligence.

## C.6. Qualitative Results of Spatial Understanding

We present qualitative examples of SpatioLM on VSI-Bench in Fig. 13, spanning eight categories of spatial understanding tasks. These tasks cover three major types of spatial reasoning: configurational reasoning *(object count, relative distance, relative direction, route planning)*, measurement estimation *(object size, room size, absolute distance)*, and spatiotemporal reasoning *(appearance order)*.

The results demonstrate that SpatioLM exhibits strong and consistent visual spatial understanding capabilities across all task types. For configurational reasoning tasks, such as object counting, relative distance and direction estimation, and route planning, SpatioLM accurately captures the spatial structure of complex scenes, indicating an intuitive understanding of object layouts and inter-object relationships. In measurement estimation tasks, including object size, room size, and absolute distance, while precise numerical prediction remains inherently challenging, SpatioLM produces reasonable and coherent estimates, reflecting a well-calibrated internal representation of scale and distance. Furthermore, for spatio-temporal reasoning tasks such as appearance order, SpatioLM effectively maintains spatial memory across video frames and performs coherent temporal reasoning.

Overall, these qualitative results highlight SpatioLM's comprehensive visual spatial intelligence, supporting a broad range of spatially grounded reasoning tasks and embodied interaction scenarios.

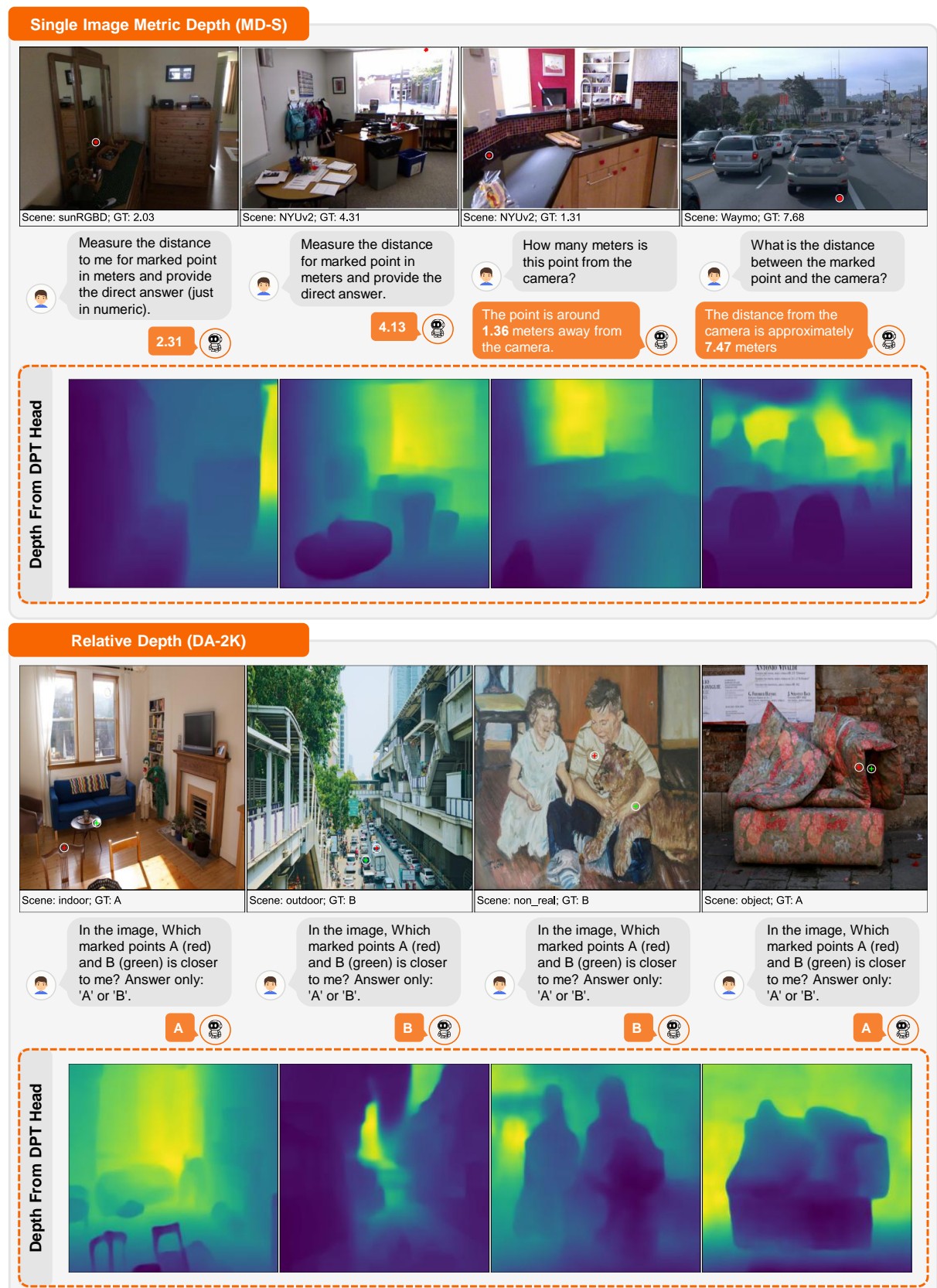

*Figure 10.* Spatial perception task visualization on MD-S and DA-2K benchmarks.

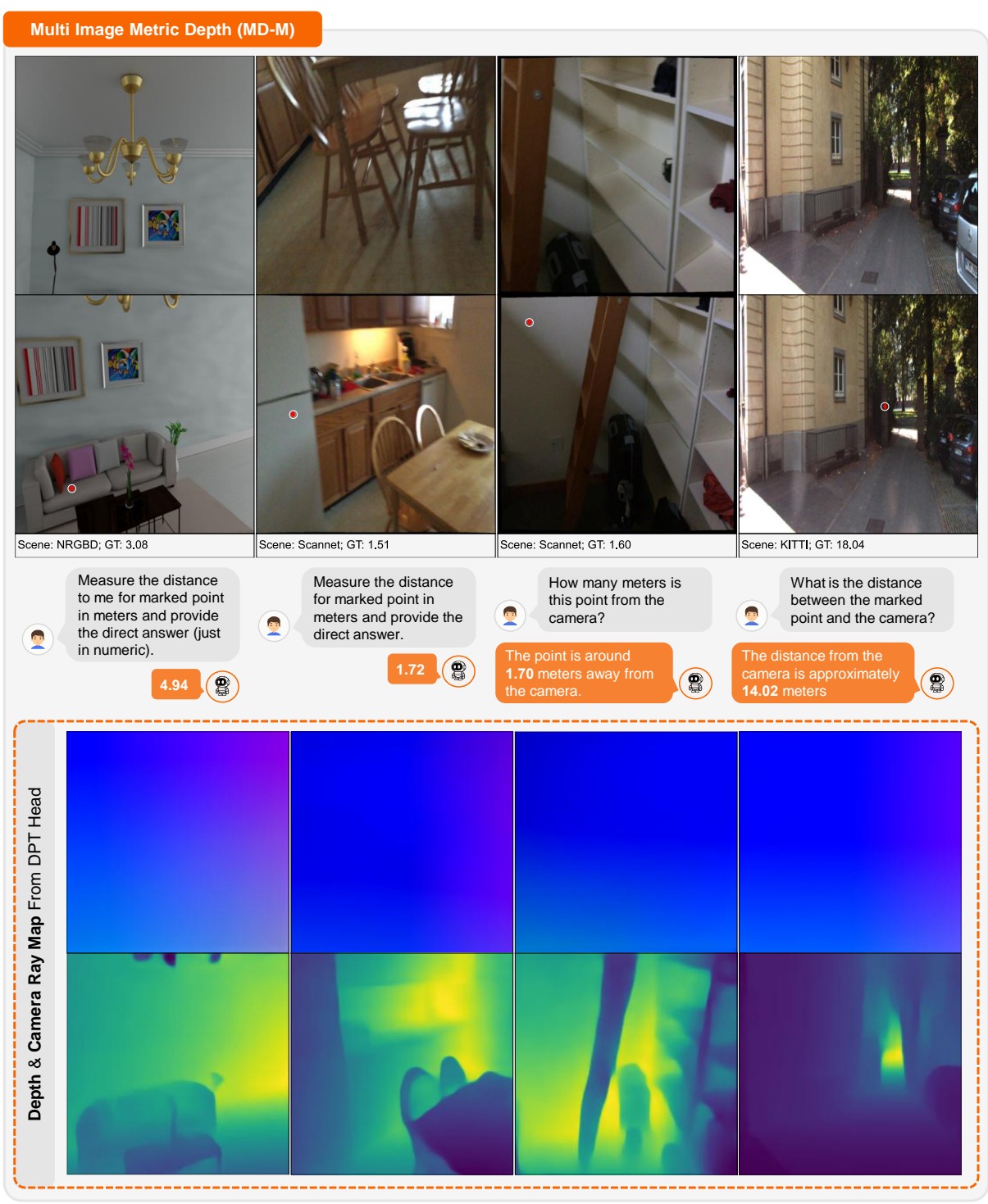

*Figure 11.* Spatial perception task visualization on MD-M benchmark.

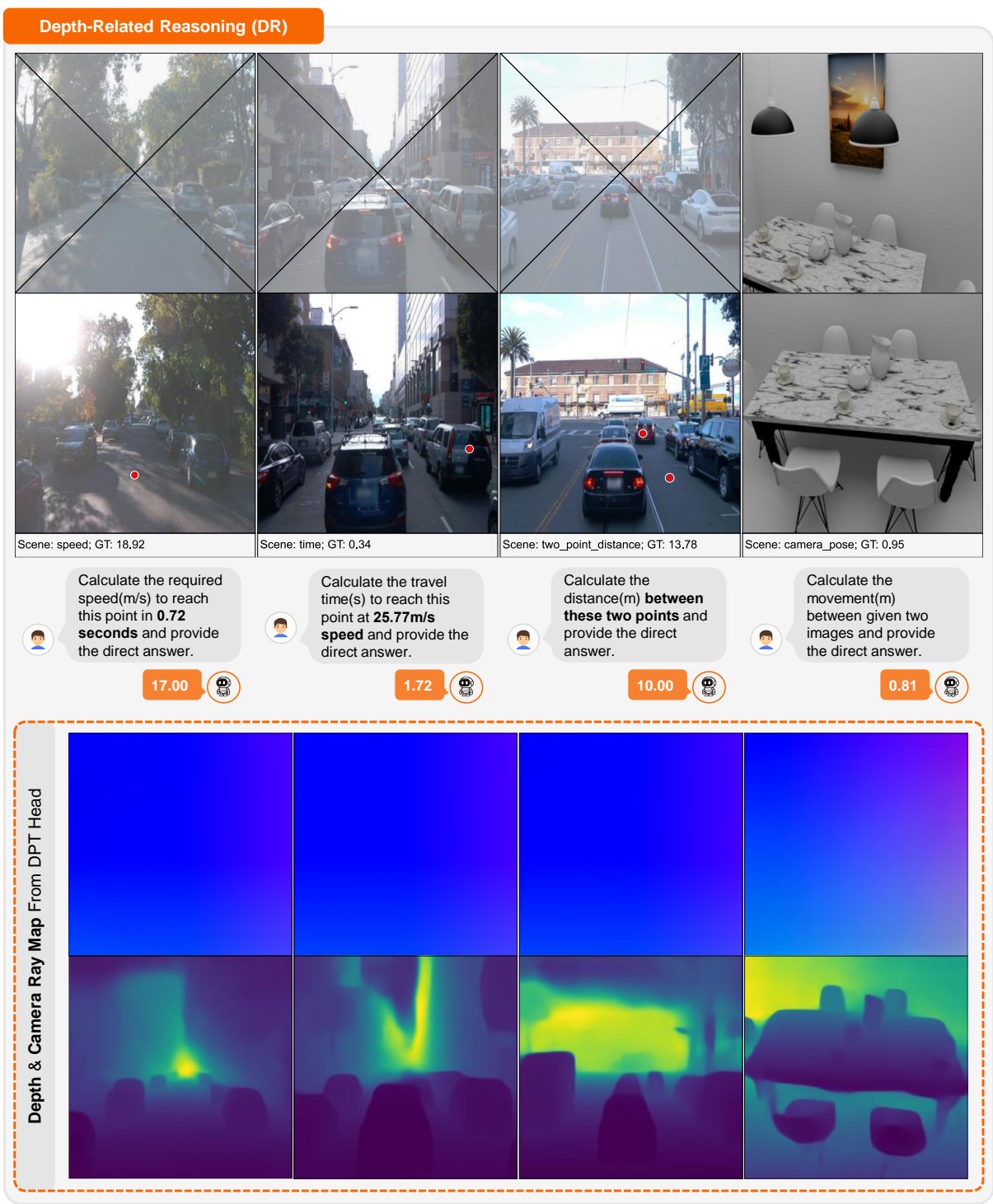

*Figure 12.* Spatial perception task visualization on DR benchmark.

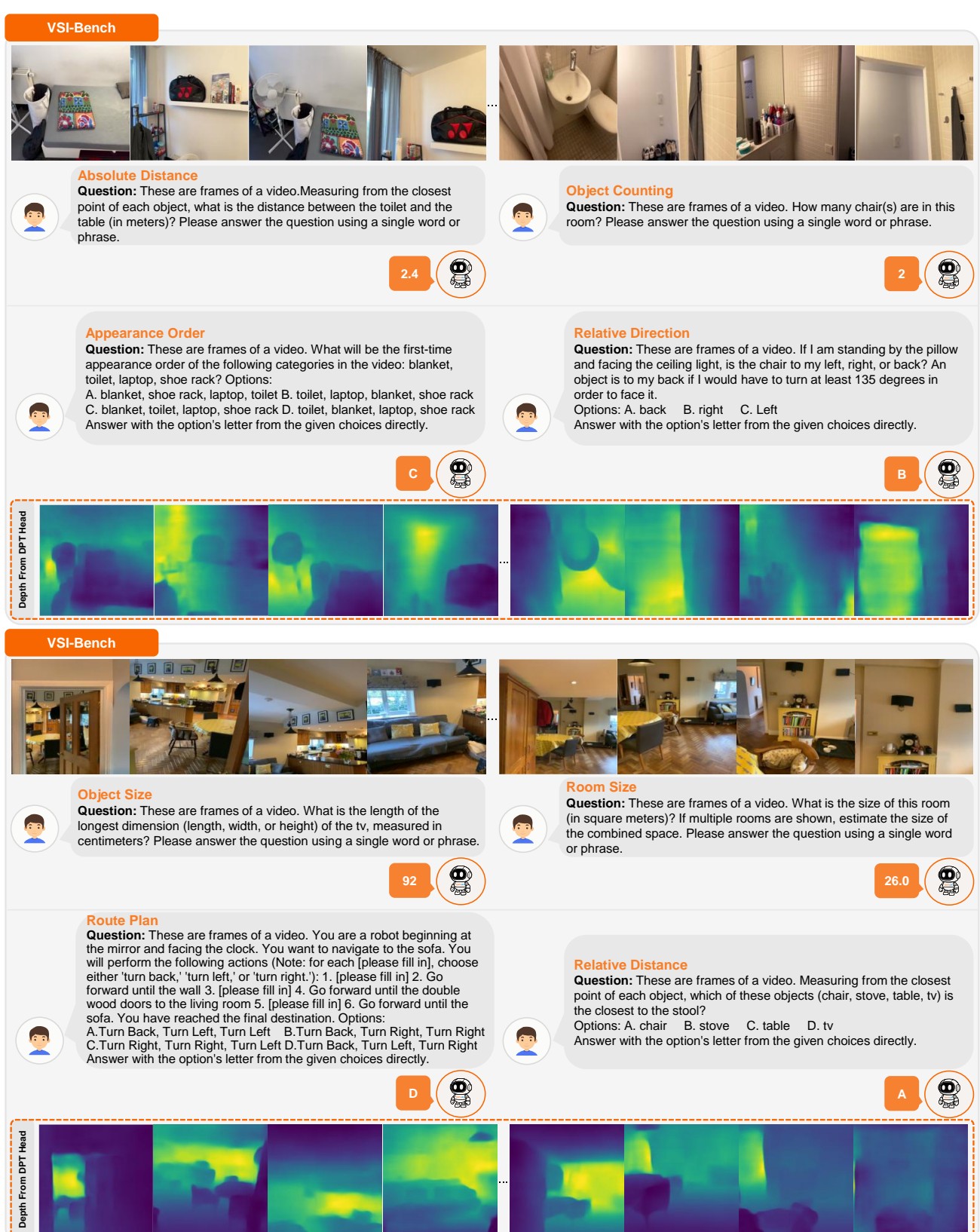

*Figure 13.* Spatial understanding task visualization on VSI-Bench.

