# OpenReview forum: "SpatioLM: Towards General Physical Spatial Intelligence in Vision-Language Models"
_ICML.cc/2026/Conference — ICML 2026 spotlight_

### Official Review · Reviewer_psLy · 2026-02-19

**Soundness:** 2
**Presentation:** 4
**Significance:** 3
**Originality:** 3
**Overall Recommendation:** 5
**Confidence:** 4

**Summary:**

The authors present SpatioLM, a framework that improves spatial reasoning in 3D without relying on 3D inputs or external encoders. SpatioLM is trained to elicit the 3D-awareness in VLMs and produce features aligned with a 3D-aware teacher DepthAnythingV3. The authors evaluate this framework on spatial reasoning, depth prediction, and robotic manipulation.

**Compliance With Llm Reviewing Policy:**

Affirmed.

**Final Justification:**

My only original concerns were with data leakage in evaluation and relevant literature being omitted. The authors have addressed both of these concerns, I feel more confident in their evaluations now that they have clarified the dataset splits used in training/evaluation. I am happy to raise my score to "Accept".

**Key Questions For Authors:**

I do not have any isolated questions, please refer to the weaknesses for my main concerns. Specifically regarding VSI-Bench, omitted relevant works, and language surrounding claims.

**Limitations:**

There is no discussion of limitations. I believe this point ties in to my last stated Weakness. In general, I feel the authors may have overstated some claims and I advise them to revise some of the language. Also, it would be interesting to see which examples the method fails on, and whether these are consistent with the baselines. Lastly, an analysis on why performance decreases on general video reasoning benchmarks (e.g. MVBench) is necessary.

**Strengths And Weaknesses:**

**Strengths**
- The approach is novel and addresses issues in prior works integrating 3D-aware features into the LLM decoding process.
- The figures are clear and well-presented. Generally the paper is well-written and I had an enjoyable time reading it.
- The experiments are very thorough, the authors took great care with the ablations, which I greatly appreciate. Though, I have a large concern regarding VSI-Bench (see Weaknesses).

**Weaknesses**
- My primary concern is that the main dataset used to evaluate improved spatial reasoning is VSI-Bench. However, VSI-Bench is composed of data from (1) ScanNet, (2) ScanNet++, (3) ARKitScenes. All three of these datasets are included in the Training data of SpatioLM (Table 7). Thus, I must conclude that there is significant data-leakage in the evaluation setting, which makes it impossible to evaluate the benefit of SpatioLM. Moreover, it is extremely concerning that details of the training corpus are not discussed in the main text (**L298-302**). I recommend the authors clarify this issue of data leakage and repeat their experiments without using these corpuses for training. Otherwise, I cannot reasonably evaluate the performance of SpatioLM on VSI-Bench. This is the reason behind my "Soundness" rating.
- The paper claims there are "two categories" of work to provide VLMs the ability to reason in 3D. However, this is missing an extensive line of work on *tool-use*. This is a major omission in the Related Works and the comparisons. Specifically, the authors highlight that their method does not require "geometric inputs" **L125** and do not require "invasive 3D data fine-tuning" **L132** - but these are exactly the issues that methods using tool-use circumvent. In this body of work, there are inference-only approaches that do not require any training data (3D or otherwise) [1 - 3] and additional methods that do not train with 3D supervision [5, 6]. A discussion (or quantitative comparison) of the two approaches is necessary.
- I recommend the authors to revisit the language surrounding some of their claims. For example, it seems inappropriate to say that SpatioLM "retains favorable generalization performance" and "maintains general-purpose capabilities" when the SpatioLM instance with Intern3.5-VL suffers a 17% decrease on MVBench and 12% decrease on VideoMME. There is no harm in saying that the decrease in performance is minimal, but at the moment the claims are slightly overstated.


I am happy to increase my score should the authors sufficiently address my stated issues.



[1] - "Vipergpt: Visual inference via python execution for reasoning." Proceedings of the IEEE/CVF international conference on computer vision. 2023.

[2] - "Visual agentic ai for spatial reasoning with a dynamic api." Proceedings of the Computer Vision and Pattern Recognition Conference. 2025.

[3] - "Visual programming: Compositional visual reasoning without training." Proceedings of the IEEE/CVF conference on computer vision and pattern recognition. 2023.

[4] - "Grounded reinforcement learning for visual reasoning." arXiv preprint arXiv:2505.23678 (2025).

[5] - "No Labels, No Problem: Training Visual Reasoners with Multimodal Verifiers." arXiv preprint arXiv:2512.08889 (2025).

---

> ### Author Rebuttal · Authors · 2026-03-30
>
> We deeply appreciate the reviewer's dedicated time and are highly encouraged that the reviewer **enjoyed reading our paper, finding our approach novel and our ablations thorough**. We address the specific weaknesses (W) and limitations (L) raised below:
>
> ---
>
> ### W1: Potential data leakage — VSI-Bench. Training corpus details missing from main text.
>
> We appreciate the concern. We clarify that there is **absolutely no data leakage**. While both share the same source datasets, they strictly use **mutually exclusive data splits (Train vs. Val/Test)** at the scene level.
>
> **① Dataset Protocol Verification.** VSI-Bench [1] sources evaluation videos exclusively from *validation sets*. Our training datasets [2-4] strictly adhere to *training splits*. For instance, the VSI-590K [4] authors explicitly noted: *"we only use the training split, while the benchmarks use validation and test splits."*
>
> **② Scene-level Cross-check.** We conducted an exhaustive cross-check of all unique *scene_id*s and **confirm exactly zero overlap** between our training corpus and VSI-Bench. The verification result is at [`anonymous link`](https://anonymous.4open.science/api/repo/spatiolm-2702/file/data_check-mvbench.pdf).
>
> **③ Transparency.** We have added training corpus details and split isolation strategy to the revision.
>
> [1] Thinking in space: How multimodal large language models see, remember, and recall spaces. CVPR 2025.
>
> [2] Scanqa: 3d question answering for spatial scene understanding. CVPR 2022.
>
> [3] SQA3D: Situated Question Answering in 3D Scenes. ICLR 2023.
>
> [4] Cambrian-S: Towards Spatial Supersensing in Video. ICLR 2026.
>
> ---
>
> ### W2: Omission of tool-use paradigm in Related Work.
>
> We thank for your highlighting the "Tool-use" paradigm [1-5]. We have revised **Related Work** to categorize 3D-reasoning VLMs into **three** paradigms, and clarified how SpatioLM provides a complementary solution:
>
> > *"**Tool-use and Agentic Spatial Reasoning.** A prominent line of work addresses 3D reasoning by leveraging LLMs to call external APIs (e.g., depth estimators, 3D detectors) [1-3], or by utilizing RL/verifiers without explicit 3D supervision [4, 5]..."*
>
> We also expanded our Conclusion to highlight that foundation models and tool-use agents are synergistic: **a spatially-aware foundation model serves as a more reliable backbone for agentic systems, while tool-use frameworks maximize reasoning ceilings for complex tasks.**
>
> Fair quantitative comparison within the rebuttal period is challenging due to unstandardized APIs and distinct prompt strategies. We have added a qualitative comparison in the Appendix.
>
> ---
>
> ### W3: Claims about preserving general capabilities are overstated.
>
> We appreciate this feedback. Our original intent was to highlight that SpatioLM's average performance degradation on general tasks is constrained within 5%. To accurately reflect this and avoid overstatement, we have revised manuscript. Accordingly, 'maintains general-purpose capabilities' in the Abstract has been updated to 'effectively limits the degradation of general capabilities', and 'retains favorable generalization performance' in the Contributions is revised to 'mitigates the decline of general-purpose capabilities'.
>
> ---
>
> ### L: Missing limitations section;  failure case and MVBench degradation analysis.
>
> **① Failure Case Analysis.**
> We conducted a systematic failure analysis on VSI-Bench (5,130 samples).
>
> **Table A: Failure Consistency**
>
> | Task Type | Ours Fail | Baseline Fail | Shared | Consistency |
> |---|---|---|---|---|
> | Route Plan | 101 | 100 | 96 | **95.0%** |
> | Rel. Dir. | 120 | 166 | 64 | 53.3% |
>
> Shared failures concentrate on route plan (95% overlap), which demands long-horizon memory and shifting spatial references—a limitation inherited from the base model rather than caused by the SV-Module. In contrast, our method clearly outperforms the base model on relative direction (53.3% unique base failures), confirming effective spatial reasoning enhancement.
>
> **② MVBench Analysis.**
>
> **Table B: Top-3 Degraded vs. Top-2 Preserved Subtasks**
>
> | | Subtask | Δ InternVL | Δ SenseNova |
> |---|---|---|---|
> | ↓1 | Action Count | −35.5 | +5.0 |
> | ↓2 | Moving Direction | −30.5 | −25.0 |
> | ↓3 | Action Antonym | −28.0 | −28.0 |
> | ↑2 | Egocentric Navigation | +1.5 | −5.0 |
> | ↑1 | Episodic Reasoning | +3.0 | +0.5 |
>
> The degradation is highly non-uniform: *motion/dynamics tasks suffer most* because geometric supervision emphasizes static spatial structure, creating competition with temporal cues (`Details linked in W1`). *Scene-level reasoning is preserved or improved*, as it aligns with 3D spatial understanding.
>
> **③ Limitation Statement.**
> We have added a Limitations section: (1) *Pseudo-label dependence:* performance is upper-bounded by teacher model accuracy;  (2) *Motion-dynamics trade-off:* as MVBench analysis reveals, geometric supervision partially trades temporal-dynamic capacity—supplementing with temporal signals is a clear future direction.

---

> > ### Author Rebuttal · Reviewer_psLy · 2026-04-02
> >
> > I thank the authors for the comments. They have addressed my core concerns of data leakage, omissions in related works, and paper positioning. I am happy to revise my score to "Accept" if the authors include all the aforementioned things in their revised manuscript. I congratulate the authors for a strong rebuttal.

---

> > > ### Author Response · Authors · 2026-04-03
> > >
> > > Thank you sincerely for your positive acknowledgement and for taking the time to provide such constructive feedback throughout the review process. Your insightful comments on data leakage, tool-use paradigm coverage, claim calibration, and limitations have substantially improved the quality of our manuscript.
> > >
> > > We have incorporated **all** the requested changes into the revised manuscript. However, as the submission platform currently does not support updating the draft at this stage, we have prepared a detailed revision summary documenting each modification with the corresponding manuscript screenshots. It is available at the following anonymous link:[`Revision Summary (PDF)`](https://anonymous.4open.science/api/repo/spatiolm-2702/file/Revision_Summary_Rv4.pdf)
> > >
> > > We hope this summary clearly demonstrates that every concern raised has been faithfully addressed, consistent with the conditions you outlined for revising your recommendation.
> > >
> > > **Should you have any additional questions or concerns, we would be very happy to address them promptly.**
> > >
> > > Thank you again for your valuable guidance that has meaningfully strengthened this work.

---

### Official Review · Reviewer_eYZ9 · 2026-03-06

**Soundness:** 3
**Presentation:** 3
**Significance:** 3
**Originality:** 2
**Overall Recommendation:** 5
**Confidence:** 4

**Summary:**

This paper proposes **SpatioLM**, a frozen-backbone VLM with a plug-and-play spatio-vision side module injected into LM blocks via zero-initialized projections, trained with language loss plus pseudo depth / ray-map supervision and feature distillation. It reports gains on VSI-Bench, ScanQA, SQA3D, several depth-centric benchmarks, and a LIBERO transfer setting.

**Compliance With Llm Reviewing Policy:**

Affirmed.

**Final Justification:**

This paper features extensive experiments and is very solid, demonstrating its effectiveness across multiple domains. I strongly support its acceptance.

**Key Questions For Authors:**

1. I am curious whether unfreezing the VLM—either through full parameter updates or LoRA—would lead to superior results compared to the current frozen baseline.

2. “No noticeable degradation” in general ability is too strong: the averages are small, but some absolute drops are not, e.g. InternVL3.5 falls from 73.2 to 60.5 on MVBench and from 65.2 to 57.7 on VideoMME.

**Limitations:**

See above.

**Strengths And Weaknesses:**

## Strengths

1. The frozen-backbone, side-tuning design is practically attractive and clearly motivated. It is simpler than dual-encoder geometry systems and is easy to attach or remove at inference.

2. The empirical coverage is broad: spatial perception, spatial understanding, general-capability retention, and manipulation transfer are all tested. This is stronger than a narrow benchmark paper.

## Weaknesses

1. **Details and hyperparameters are insufficiently specified**: What is the exact length of the historical frames? Which specific intermediate layer of the vision encoder is selected for token extraction? Furthermore, how exactly are the tokens chosen in the "Select Vision Tokens" process illustrated in Figure 3? Additionally, the Spatio-Vision Block emphasized in the paper closely resembles an **adapter** design; what are its specific model size and parameter count? These technical details should be explicitly articulated in the manuscript, ideally supported by corresponding ablation studies.

2. **Missing Core Architectural Ablations**: The authors highlight the "Alternating-Attention mechanism" (Frame and Global Attention) as the core mechanism to capture intra-frame and inter-frame geometric consistency. Yet, Table 5 and Table 6 completely lack an ablation study validating this specific module against standard self-attention baselines.

3. **Novelty is weaker than claimed**: Recent work already studies 2D-only spatial enhancement for VLMs/VLAs via geometry-aware augmentation or dedicated spatial tuning, including Spatial-MLLM, VG-LLM, VLM-3R, VST, and SpatialForcing. The real delta here is a specific adapterized side-path plus geometric supervision.

---

> ### Author Rebuttal · Authors · 2026-03-30
>
> We sincerely thank the reviewer for the constructive feedback and for recognizing **the practically attractive design and the broad empirical coverage** of our work. We address the specific weaknesses (W) and questions (Q) raised below:
>
> ---
>
> ### W1: Missing technical details — frame length, ViT layer choice, token selection process, SV-Block parameter count.
>
> We thank the reviewer for this important question. We provide the requested details:
>
> **① Historical Frames.** As stated in the Appendix (L827-L829), image data are resized to 448×448, while video data are uniformly sampled into **32 frames** before applying the same resolution normalization.
>
> **② Intermediate Layer Selection.** Recent work [1] shows intermediate ViT layers contain richer spatial information than the final layer for geometric tasks. We select the **16th** layer from the 24-layer ViT, guided by analysis showing layers around 16/24 yield the strongest spatial embeddings (Figure 8 in [1]).
>
> **③ "Select Vision Tokens" Process.** The formulation   $H_{j\-1}^{v} = {H}_{j-1}^{m}[\mathcal{I}^{v}]$  extracts vision tokens by the index set $\mathcal{I}^{v}$, which corresponds to **all vision tokens** at the current layer, clearly distinguished from linguistic token indices.
>
> **④ SV-Block Size.** Each SV-Block contains **~50M** parameters. As shown in **Table 6**, the ablation demonstrates that **6 SV-Blocks** (totaling only **0.3B** trainable parameters) achieves optimal performance, balancing capacity and efficiency.
>
> These details have been added explicitly in the revised manuscript.
>
> [1] Perception Encoder: The best visual embeddings are not at the output of the network. NeurIPS 2025.
>
> ---
>
> ### W2: Missing Architectural Ablations (Alternating-Attention vs Self-Attention).
>
> We appreciate this critical observation. We have conducted the requested ablation:
>
> | Attention Mechanism | VSI-Bench | ScanQA CIDEr | SQA3D EM-R1 |
> | :--- | :---: | :---: | :---: |
> | Self-Attention | 70.4 | 98.7 | 56.8 |
> | **Alternating-Attention (Ours)** | **71.6** | **101.8** | **58.6** |
>
> Our Alternating-Attention consistently outperforms the self-attention baseline (+1.2 on VSI-Bench, +3.1 on ScanQA, +1.8 on SQA3D), validating that explicitly decoupling intra-frame and inter-frame attention is beneficial for capturing geometric consistency. This ablation has been added to the revision.
>
> ---
>
> ### W3: Novelty is an adapterized side-path + geometric supervision; differentiate from concurrent works.
>
> We thank the reviewer for highlighting these concurrent works. We appreciate this characterization of our design. Beyond the architectural form, the key distinction is: **SpatioLM does not rely on any external 3D priors, but instead elicits latent 3D spatial capabilities already present within the pretrained VLM's vision encoder.** This insight—that rich spatial knowledge can be elicited from 2D vision tokens without heavy external encoders—represents a distinct paradigm from Spatial-MLLM, VG-LLM, and VLM-3R, which all depend on external spatial encoders.
>
> | Method | No External Encoders | Architectural Spatial Design | Geometric Supervision | Preserves VLM Generality |
> | :--- | :---: | :---: | :---: | :---: |
> | Spatial-MLLM/VG-LLM/VLM-3R | ✗ | ✓ | ✗ | ✗ |
> | VST | ✓ | ✗ (Data-centric) | ✗ | ✓ |
> | SpatialForcing | ✓ | ✗ | ✓ | ✗ (Action-specific)|
> | **SpatioLM (Ours)**| **✓** | **✓ (Side-path)** | **✓** | **✓** |
>
> Rather than relying on external encoders or purely data-centric tuning, our side-path efficiently elicits implicit 3D geometry while strictly preserving VLM generality. This discussion has been added to the revision.
>
> ---
>
> ### Q1: LoRA compares to the current frozen baseline.
>
> Thank you for this insightful question. We conducted the comparison using LoRA (rank=16, alpha=64) with a comparable parameter budget to our SV-Module:
>
> | Method | VSI-Bench | ScanQA CIDEr | SQA3D EM-R1 |
> | :--- | :---: | :---: | :---: |
> | LoRA (rank=16, alpha=64) | 69.8 | 97.2 | 55.7 |
> | **SpatioLM (Ours)** | **71.6** | **101.8** | **58.6** |
>
> Under a comparable parameter budget, our non-invasive SV-Module outperforms LoRA across all benchmarks (+1.8 on VSI-Bench, +4.6 on ScanQA, +2.9 on SQA3D), suggesting that a dedicated side-path design is more effective than directly fine-tuning the backbone with parameter-efficient methods. This comparison has been added to the revision.
>
> ---
>
> ### Q2: "No noticeable degradation" in general ability is too strong.
>
> We appreciate this careful observation. In our general capability evaluations, the average performance drop across general tasks is within 5%, with clear advantages over baselines. Nonetheless, to improve precision, we have revised the statement:
>
> - **Original:** *"SpatioLM does not suffer noticeable degradation in general capabilities."*
> - **Revised:** *"The performance degradation of SpatioLM on general tasks is considerably smaller than that of other baseline models."*
>
> We appreciate the reviewer's help in improving the precision of our claims.

---

> > ### Author Rebuttal · Reviewer_eYZ9 · 2026-04-03
> >
> > My concerns have been solved. I will keep my positive score.

---

> > > ### Author Response · Authors · 2026-04-05
> > >
> > > Thank you very much for your acknowledgement. We are glad that our rebuttal has addressed your concerns.
> > >
> > > We have revised the manuscript accordingly, and we will make sure these updates are reflected in the final version once the camera-ready submission opens.

---

### Official Review · Reviewer_Jptm · 2026-03-11

**Soundness:** 4
**Presentation:** 4
**Significance:** 3
**Originality:** 4
**Overall Recommendation:** 5
**Confidence:** 4

**Summary:**

The paper proposes SpatioLM, a parameter-efficient framework designed to enhance the
spatial intelligence of Vision-Language Models (VLMs). Unlike existing methods that rely on
explicit 3D priors (e.g., depth maps) or third-party spatial encoders, SpatioLM uses a plugand-
play Spatio-Vision Module (SV-Module) to elicit implicit geometric knowledge directly
from the pretrained VLM's vision tokens. The model is trained using pseudo-depth and
camera ray supervision to guide the learning of physically coherent representations. It
achieves state-of-the-art performance on spatial perception and understanding benchmarks
(e.g., VSI-Bench) while maintaining the general-purpose capabilities of the base VLM.

**Compliance With Llm Reviewing Policy:**

Affirmed.

**Final Justification:**

This paper presents SpatioLM, a parameter-efficient framework that enhances spatial intelligence in VLMs through a novel plug-and-play SV-Module, eliminating the need for explicit 3D inputs at inference time. The empirical evaluation is comprehensive and demonstrates strong performance across a wide range of benchmarks. The rebuttal also addresses my concerns. Overall, I recommend acceptance of this paper.

**Key Questions For Authors:**

1. Please clarify whether the claim of “without extra 3D priors” only means that no explicit
3D inputs are required at inference time. Given that the training process uses pseudo
depth, ray maps, camera supervision, and a Depth Anything V3 teacher, does this not
still amount to using external geometric priors to train the model?
2. What experimental evidence supports the claim that the SV-Module “elicits” spatial
knowledge already present in the backbone, rather than simply learning spatial
capabilities anew through geometric supervision in the new module?

**Limitations:**

No.

The authors should more explicitly discuss:
1. The methodological limitation that the approach still depends on external geometric
supervision during training, even though it emphasizes the absence of extra 3D priors
at inference time.
2. The model may produce unreliable metric or spatial judgments in safety-critical settings
such as robotics and autonomous driving. Overestimating the spatial reliability of such
systems in real-world embodied or decision-making applications may introduce risks.

**Strengths And Weaknesses:**

Strengths
1. The proposed Plug-and-Play SV-Module paradigm is innovative, effectively eliciting 3D
geometric structures from pretrained tokens without requiring auxiliary 3D input at
inference time.
2. The design of the Dual DPT Head, which predicts dense depth maps and ray maps,
provides explicit geometric supervision during the training phase.
3. SpatioLM models diverse spatial tasks—ranging from spatial perception to spatial
understanding—as conditional language generation tasks within a unified task
formulation, thereby simplifying the inference workflow.
4. The evaluation is extensive, spanning low-level spatial perception (MD-S, MD-M, DA-2K,
DR), high-level spatial understanding (VSI-Bench, ScanQA, SQA3D), and embodied
manipulation tasks (LIBERO).

Weaknesses
1. SpatioLM's performance may be bottlenecked by the teacher model's limitations in
handling complex structures and reflective surfaces due to noisy pseudo-labels.
The evaluation of general capabilities is not fully controlled. The Spatial-MLLM baseline
uses a different VLM backbone from the proposed method (Qwen2.5-VL-3B versus
InternVL3.5-8B / SenseNovaSI-8B), and the paper does not clearly establish alignment in
training setup. So the performance differences on VideoMME, VideoMMMU, MVBench,
and MMMU may reflect backbone and training discrepancies rather than the effect of
the proposed spatial module alone.
2. The authors should clarify that "pure-vision" refers only to the lack of dedicated 3D
sensors, as the framework still heavily relies on language-based instruction tuning.
3. The lack of explanation for the fixed loss weights (α=0.6,β=0.2,γ=0.2). Will the model be
sensitive to these hyperparameters?

---

> ### Author Rebuttal · Authors · 2026-03-30
>
> We are grateful for the reviewer’s thoughtful critique and for highlighting **the innovativeness of our SV-Module and the extensive evaluation across diverse tasks**. We address the specific weaknesses (W), questions (Q) and limitations (L) raised below:
>
> ---
>
> ### W1: ① Noisy pseudo-labels may bottleneck performance; ② Baseline uses different backbone.
>
> **①** We acknowledge that pseudo-label noise can limit the performance ceiling. However, pseudo-labels enable scalable training at substantially lower cost than manual 3D annotation. Importantly, our SV-Module is fully compatible with ground-truth annotations when available.
>
> **②** We present controlled comparisons on the **same SenseNovaSI-8B backbone**:
>
> **Table A: General Capabilities:**
>
> | Model | Avg | VideoMME | MVBench | MMMU | VMMU-P/C/A |
> | :--- | :---: | :---: | :---: | :---: | :---: |
> | SenseNovaSI-8B | 49.8 | 62.5 | 65.6 | 50.6 | 53.0/36.3/30.7 |
> | Spatial-MLLM | 42.7 (-14%) | 52.6 | 60.4 | 51.1 | 35.7/28.0/28.7 |
> | **SpatioLM** | **48.1 (-3%)** | 60.7 | 60.7 | 50.1 | 51.7/33.7/31.7 |
>
> **Table B: Spatial Reasoning (VSI-Bench):**
>
> | Model | Avg |
> | :--- | :---: |
> | SenseNovaSI-8B | 68.7 |
> | Spatial-MLLM | 65.1 |
> | **SpatioLM** | **71.6** |
>
> On the same backbone, SpatioLM achieves superior spatial reasoning (+2.9 vs. -3.6) and much less general degradation (-3% vs. -14%), confirming gains stem from our SV-Module, not backbone differences.
>
> ---
>
> ### W2: "Pure-vision" may be ambiguous given reliance on language instruction tuning.
>
> We appreciate this observation. Since the term appeared **only once**, we have revised it directly:
> *   **Original:** "...a *pure-vision* framework..."
> *   **Revised:** "...a ***purely 2D vision-language*** framework..."
>
> ---
>
> ### W3: Lack of explanation for fixed loss weights (α=0.6, β=0.2, γ=0.2) and sensitivity.
>
> Thank you for this important question. Inspired by prior works on geometric supervision (3DRS [1] and Video-3D-LLM [2]), we initially set the loss weights empirically (α=0.6, β=0.2, γ=0.2) as a reasonable starting point. Following your valuable suggestion, we conducted a systematic ablation:
>
> | α | β | γ | VSI-Bench | ScanQA CIDEr | SQA3D EM-R1 |
> | :---: | :---: | :---: | :---: | :---: | :---: |
> | 0.4 | 0.2 | 0.2 | 71.5 | 100.2 | 58.8 |
> | **0.6** | **0.2** | **0.2** | 71.6 | 101.8 | 58.6 |
> | 0.8 | 0.2 | 0.2 | **71.9** | 101.2 | 59.9 |
> | 0.6 | 0.3 | 0.2 | 71.4 | 101.7 | 59.2 |
> | 0.6 | 0.1 | 0.2 | 71.8 | **103.2** | **60.2** |
> | 0.6 | 0.2 | 0.1 | 71.4 | 100.9 | 60.1 |
> | 0.6 | 0.2 | 0.3 | 71.7 | 102.2 | 58.5 |
> | 0.8 | 0.2 | 0.3 | **71.9** | 102.9 | 59.4 |
>
> The model is **not sensitive** to these hyperparameters—VSI-Bench varies within only 0.5 points across all configurations. We have updated the manuscript with this ablation.
>
> [1] 3DRS: MLLMs Need 3D-Aware Representation Supervision for Scene Understanding. NeurIPS 2025.
>
> [2] Video-3D LLM: Learning Position-Aware Video Representation for 3D Scene Understanding. CVPR 2025.
>
> ---
>
> ### Q1: Pseudo depth / teacher model in training — contradicts "without extra 3D priors"?
>
> We acknowledge that using a teacher model or pseudo-depth means training does leverage external geometric knowledge. The key architectural distinction is:
>
> 1. **Strictly 2D Inputs:** No explicit 3D inputs during training or inference forward passes.
> 2. **Priors as Supervision Only:** 3D priors are confined to automated pseudo-labels for loss calculation, eliminating the need for physical 3D sensors or costly annotations.
>
> We have revised accordingly: *"SpatioLM eliminates the reliance on explicit 3D prior **inputs**, distilling implicit 3D knowledge solely through automated **supervision signals** during training."*
>
> ---
>
> ### Q2: Evidence that SV-Module elicits existing spatial knowledge vs. learning from scratch?
>
> **① Frozen Backbone Design.** Our entire VLM backbone is **frozen** during training—the SV-Module operates as an external side-path reading intermediate visual features. Since backbone weights remain unchanged, it must extract spatial information already encoded in frozen vision tokens, rather than learning it anew.
>
> **② Converging Literature.** Recent studies (e.g., Perception Encoder [3]) independently confirm that pretrained vision encoders contain substantial latent spatial capabilities in intermediate layers, supporting that a dedicated module can *elicit* rather than *create* this capability.
>
> [3] Perception Encoder: The best visual embeddings are not at the output of the network. NeurIPS 2025.
>
> ---
>
> ### L1: Dependence on external geometric supervision. L2: Unreliable in safety-critical settings.
>
> **L1:** Training relies on pseudo-labels from Depth Anything V3, whose noise imposes an upper bound on 3D comprehension. This is noted in the revised manuscript.
>
> **L2:** We acknowledge SpatioLM may be unreliable for safety-critical applications and caution against such deployment without additional verification. This caveat has been added to the Limitations section.

---

> > ### Author Rebuttal · Reviewer_Jptm · 2026-04-04
> >
> > The authors have well addressed the weaknesses and answered additional questions. My concerns have been adequately addressed.

---

> > > ### Author Response · Authors · 2026-04-05
> > >
> > > Thank you very much for your acknowledgement and positive feedback. We are pleased that our rebuttal has adequately addressed your concerns and additional questions.
> > >
> > > We have made the corresponding revisions to the manuscript and will ensure that these updates are included in the final version when the camera-ready submission becomes available.

---

### Official Review · Reviewer_8hrK · 2026-03-12

**Soundness:** 4
**Presentation:** 3
**Significance:** 4
**Originality:** 3
**Overall Recommendation:** 5
**Confidence:** 4

**Summary:**

The paper introduces SpatioLM, a parameter-efficient framework that enhances physical spatial intelligence in VLMs without relying on extra 3D priors or external spatial encoders. The authors introduce a plug and play, non-invasive spatio-vision module that elicits spatial knowledge in VLM inherently. Experiments demonstrate that SpatioLM is not only strong at spatial perception and attains competitive performance when transferred to embodied manipulation tasks.

**Compliance With Llm Reviewing Policy:**

Affirmed.

**Key Questions For Authors:**

Please refer to above

**Limitations:**

Yes

**Strengths And Weaknesses:**

Strengths:
1. Strong performance. SpatialLM outperforms the strong commercial model Gemini-2.5-pro and other spatial models with only 8B parameters. In addition, Figure6 shows that the overall performance on general multimodal benchmarks are not suffered and even improves on some video benchmarks.
2. Downstream task demonstration. Section 4.4 shows that SpatioLM is also a strong foundation model for embodied manipulation adaptation.
3. The non-invasive design makes the model easier to retain the general multimodal ability, which can be potentially expanded to other domains.

Weakness:
1. It is encouraged to detailedly discuss the similarity between the training dataset and the evaluations. From the result of Table1, the huge performance divergence of different models (e.g. near zero performance of Cambrian-S-7B) might suggest the compared models lack a close-domain training dataset instead of having a flaw in model design.
2. Some presentations can be improved. For example, the radar plots in Figure1 are all not very visible.

---

> ### Author Rebuttal · Authors · 2026-03-30
>
> We deeply appreciate the reviewer's insightful comments and their positive evaluation of our **strong performance, downstream adaptability, and non-invasive design**. We address the specific weaknesses (W) raised below:
>
> ---
>
> ### W1: Training/evaluation data overlap concern; performance gap may reflect data advantage rather than model design.
>
> Thank you for the constructive feedback. While disentangling training data from model design is crucial, we respectfully clarify that **the observed performance divergence primarily stems from superior model design and task formulation, not an unfair close-domain data advantage**. We support this from three perspectives: **① Strong Zero-Shot Generalization on Unseen Domains**; **② Baseline Underperformance Stems from Task Misalignment, Not Domain Gap**; and **③ Controlled Comparison under Identical Task Formulation**.
>
> **① Strong Zero-Shot Generalization on Unseen Domains.**
> Our model avoids close-domain memorization. Among 6 evaluation datasets, 4 (**SunRGBD, NYU-v2, NRGBD, and KITTI**) are strictly unseen during training. As shown in **Table A**, SpatioLM exhibits strong zero-shot performance on these out-of-domain benchmarks, proving robust generalization over in-domain overfitting.
>
> **Table A: Depth Evaluation Setting and Zero-Shot performance of SpatioLM (SenseNovaSI-8B)**
>
> | Evaluation Dataset | Domain Status | MD-S | MD-M |
> | :--- | :---: | :---: | :---: |
> | **SunRGBD** | **Unseen (Zero-shot)** | 88.5 | - |
> | **NYU-v2** | **Unseen (Zero-shot)** | 84.6 | - |
> | **NRGBD** | **Unseen (Zero-shot)** | - | 47.6 |
> | **KITTI** | **Unseen (Zero-shot)** | - | 75.2 |
> | ScanNet(val) | Seen (In-domain) | - | 84.1 |
> | Waymo(val) | Seen (In-domain) | 77.5 | - |
>
> *(Note: Please refer to Appendix-Table 8 in the revised manuscript for comprehensive results.)*
>
> **② Baseline Underperformance Stems from Task Misalignment, Not Domain Gap.**
> Cambrian-S-7B's near-zero performance reflects a **task-level mismatch, not a domain gap**. **Table B** shows both models share multiple in-domain training sources. While Cambrian-S-7B excels at general spatial reasoning (e.g., multiple-choice or object-level estimation), it was never optimized for **point-level exact absolute depth regression**. Its underperformance thus stems from task granularity misalignment, not deficient scene understanding.
>
> **Table B: Comparison of Training Data and Task Formulation**
>
> | Model | Shared In-Domain Training Sources | Task Formulation & Output Format |
> | :--- | :--- | :--- |
> | Cambrian-S-7B | ScanNet, ScanNet++, ARKitScenes, Hypersim | Multiple-choice (relative relations) & Numerical estimation (object/room size) |
> | SpatioLM (Ours) | ScanNet, ScanNet++, ARKitScenes, Hypersim | Direct numerical depth prediction |
>
> As suggested, we have added this discussion to our revision: *"While Cambrian-S-7B demonstrates strong capabilities in broad spatial reasoning tasks, its suboptimal performance on our specific benchmark is attributed to task formulation differences rather than a lack of domain exposure, given the shared scenes (e.g., ScanNet, Hypersim) in both training corpora."*
>
> **③ Controlled Comparison under Identical Task Formulation.**
> To strictly isolate model design from data advantage, we compare SpatioLM with DepthLM (Pixtral-12B). DepthLM shares **the exact same task formulation** (QA-based depth estimation). Under this controlled setting with aligned tasks and larger data, our model still achieves massive gains across all metrics (**Table C**). This firmly validates our architectural superiority over data scale or domain coverage.
>
> **Table C: Controlled Comparison under Identical Task Formulation**
>
> | Model | Task Formulation | MD-S (↑) | MD-M (↑) | DR (↑) |
> | :--- | :--- | :---: | :---: | :---: |
> | DepthLM (Pixtral-12B)| Depth Estimation via QA | 18.0 | 40.0 | 11.9 |
> | SpatioLM (Ours) | Depth Estimation via QA | **83.5** | **69.0** | **41.8** |
>
> ---
>
> ### W2: Some presentations can be improved. For example, the radar plots in Figure 1 are all not very visible.
>
> We thank the reviewer for pointing this out. We have comprehensively updated the radar plots in Figure 1 to significantly enhance their visibility and clarity. The updated portion of Figure 1 can be viewed at the following [`anonymous link`](https://anonymous.4open.science/api/repo/spatiolm-2702/file/teaser_radar.jpeg?v=49b843d5). We have replaced the original figure with this improved version in the revision manuscript.

---

> > ### Author Rebuttal · Reviewer_8hrK · 2026-04-04
> >
> > Thanks for the response, my concerns are adequately addressed. Please make sure to update the next version of the paper regarding clarification of train/eval domain, and the new figure.

---

> > > ### Author Response · Authors · 2026-04-05
> > >
> > > Thank you for your positive acknowledgement and helpful reminder. We **have revised** the manuscript accordingly to clarify the train/eval domain and include the new figure, and we will ensure that these changes are reflected in the **final version once the camera-ready submission opens.**

---

### Decision · Program_Chairs · 2026-04-30

**Decision:**

Accept (spotlight)

**Comment:**

The paper introduces SpatioLM, a plug-and-play, non-invasive, and parameter-efficient framework designed to enhance the spatial intelligence of Vision-Language Models. SpatioLM elicits implicit geometric knowledge directly from the pretrained VLM’s vision tokens. The evaluations are conducted on several tasks, including Spatial Perception, Spatial Understanding, General Capabilities, and Embodied Manipulation, demonstrating consistent and exciting gains.

The authors provide a satisfactory and detailed response to the concerns raised during the review process, including potential data leakage between training and testing, noisy pseudo labels, hyperparameter analysis, architectural ablations, and full/LoRA fine-tuning. These clarifications sufficiently address the reviewers’ questions. After the rebuttal, all reviewers agree that this is a strong paper. Based on the overall quality of the work and the authors’ thorough responses, I recommend acceptance.